# Mechanistic insights into C-C coupling in electrochemical CO reduction using gold superlattices

Xiaoju Yang[1,2,6], Chao Rong [3,6], Li Zhang[1,2], Zhenkun Ye[1,2], Zhiming Wei[4], Chengdi Huang[1,2], Qiao Zhang[4], Qing Yuan[1,2], Yueming Zhai [4], Fu-Zhen Xuan [3], Bingjun Xu [5] ✉, Bowei Zhang [3] ✉ & Xuan Yang [1,2] ✉

Developing in situ/operando spectroscopic techniques with high sensitivity and reproducibility is of great importance for mechanistic investigations of surface-mediated electrochemical reactions. Herein, we report the fabrication of highly ordered rhombic gold nanocube superlattices (GNSs) as substrates for surface-enhanced infrared absorption spectroscopy (SEIRAS) with significantly enhanced SEIRA effect, which can be controlled by manipulating the randomness of GNSS. Finite difference time domain simulations reveal that the electromagnetic effect accounts for the significantly improved spectroscopic vibrations on the GNSs. In situ SEIRAS results show that the vibrations of CO on the $Cu_2O$ surfaces have been enhanced by $2.4 \pm 0.5$ and $18.0 \pm 1.3$ times using GNSs as substrates compared to those on traditional chemically deposited gold films in acidic and neutral electrolytes, respectively. Combined with isotopic labeling experiments, the reaction mechanisms for C-C coupling of CO electroreduction on Cu-based catalysts are revealed using the GNSs substrates.

The rational design of stable and high-performance materials for the storage and conversion of renewable energy demands mechanistic understanding of the involved reactions. In the past few decades, a variety of experimental techniques and simulation methods have been developed to identify the reaction intermediates and reveal the reaction pathways of surface-mediated electrochemical reactions. Among which, attenuated total reflectance–surface-enhanced infrared absorption spectroscopy (ATR-SEIRAS) has been widely utilized in studying various important electrochemical processes including methanol oxidation[1-3], hydrogen oxidation/evolution[4-6], and $CO_2$/CO electrochemical reduction[7,8]. The contribution of the spectroscopic signal decreases exponentially as the distance between the molecule and the metal surface grows, which makes it ideal to selectively detect species adsorbed on and near (<10 nm) the electrochemical interfaces. Typically, films with rough metal islands are necessary to achieve significantly enhanced vibrational signals of adsorbed species[9,10]. The dramatically changed optical properties of adsorbed molecules, known as the SEIRA effect, is strongly dependent on the morphology of metal films. Currently, most metal films for ATR-SEIRAS in catalysis applications are prepared via a wet-chemical deposition approach, which causes the lacking of control on the shapes of individual metal nanoparticles and the morphology of metal films[11-13]. The relatively poor sensitivity and reproducibility of chemically deposited metal films make it challenging to obtain the complete picture of

[1]Key Laboratory of Material Chemistry for Energy Conversion and Storage, Huazhong University of Science and Technology, Wuhan 430074, China. [2]Hubei Key Laboratory of Bioinorganic Chemistry and Materia Medica, School of Chemistry and Chemical Engineering, Huazhong University of Science and Technology, Wuhan 430074, China. [3]Shanghai Key Laboratory of Intelligent Sensing and Detection Technology, School of Mechanical and Power Engineering, East China University of Science and Technology, Shanghai 200237, China. [4]The Institute for Advanced Studies, Wuhan University, Wuhan 430072, China. [5]College of Chemistry and Molecular Engineering, Peking University, Beijing 100871, China. [6]These authors contributed equally: Xiaoju Yang, Chao Rong. ✉e-mail: b_xu@pku.edu.cn; boweiz@ecust.edu.cn; xuanyang@hust.edu.cn

mechanistic information and may lead to incomplete/incorrect mechanistic understandings.

In the past few decades, various nanofabrication technologies have been developed to precisely control the morphology and pattern of metal films on different substrates, including electron beam lithography, direct laser writing, interference lithography, and photolithography, etc[14–18]. However, lithography techniques for the fabrications of metal films typically require the utilization of expensive facilities and consist of multiple comprehensive steps. Self-assembly of nanocrystals mimics the growth of atoms into a crystal and has attracted extensive research interest, which provides an ideal approach to manipulating the pattern and morphology of metal films on a variety of substrates. Recently, close-packed superlattices has been rapidly developed, because of the successful synthesis of nanocrystals with different structures and shapes[19,20]. However, it remains challenging to fabricate ordered superlattices over long distances relative to individual nanocrystal size for device applications.

Here, we demonstrate the successful application of self-assembled gold superlattices on the surfaces of Si crystals as the substrates for in situ ATR-SEIRAS, with significantly enhanced SEIRA effect of around one order of magnitude near the electrochemical interfaces. Starting from gold nanocubes of 40 nm in edge length, we have fabricated rhombic gold nanocube superlattices (GNSs) on the surfaces of Si crystals with a diameter of 2 cm and systematically evaluated the conditions for self-assembly process to ensure high quality and good reproducibility. The SEIRA effect of GNSs is found to be highly dependent on the pattern of the film and the distance between adjacent gold nanocubes. The superlattice with an interspace of 5−8 nm shows the highest SEIRA effect with enhancements of $6.4 \pm 1.7$ and $8.1 \pm 1.7$ compared to traditional chemically deposited gold films (CDFs) in 0.1 M $HClO_4$ and 0.5 M $KHCO_3$, respectively. Finite difference time domain (FDTD) simulations reveal that the enhanced electromagnetic field leads to significantly improved spectroscopic vibrations on the GNSs. Furthermore, such gold superlattices are successfully applied as the substrates to study the electrochemical CO reduction reactions (CORR) on the $Cu_2O$ surfaces in acidic and neutral environments. The vibrational signals of adsorbed CO are enhanced by $2.4 \pm 0.5$ and $18.0 \pm 1.3$ times as compared to those on the CDFs in 0.1 M $HClO_4$ and 0.5 M $KHCO_3$, respectively. Combined with isotopic labeling experiments, SEIRAS results on the gold superlattices reveal that the coupling reactions between adsorbed CO and $CH_3$ group from $CH_3I$ yield the production of ethanol via the CORR. Therefore, our results provide a novel approach with high sensitivity and reproducibility to revealing the reaction mechanisms of surface-mediated electrochemical reactions for renewable energy applications.

## Results

### Morphology and stability of rhombic gold nanocube superlattices

The first step in the preparation of self-assembled gold films involves the synthesis of gold nanocubes by following the previously reported protocols[21]. The gold nanocubes used in this work show an average edge length of 40 nm and a single localized surface plasmon resonance (LSPR) peak at 547 nm (Supplementary Fig. 1), which confirms the small size of the nanocubes. The full width at half maximum (FWHM) of the LSPR peak was only 58 nm, indicating a narrow size distribution for the gold nanocubes. As illustrated in Fig. 1a, the uniform gold nanocubes serve as building blocks for the fabrication of self-assembled gold films (SAFs) via a vapor-evaporation procedure (details in the section of Methods)[19]. The as-prepared SAFs show a rhombic superlattice at a cetylpyridinium chloride (CPC) concentration of 5 mM, which is utilized as the substrates for further spectroscopic study of CORR. Different from traditional CDFs with an irregularly ordered structure and a thickness of 52 nm (Supplementary Fig. 2), the thickness of GNSs is a monolayer of gold nanocubes (Fig.1b and

Supplementary Fig. 3). Moreover, the electrochemical impedance spectroscopy shows the excellent conductivity of GNSs, which makes them suitable for studying electrochemical reactions (Fig. 1c and Supplementary Fig. 4).

The GNSs on the silicon crystal show much enhanced stabilities in acidic and neutral environments, with a steady resistance (~7 Ω) and negligible changes after soaking in the electrolytes for 3 h (Supplementary Fig. 5). The stability of GNSs under reaction conditions is further investigated. Both the rhombic structure of the superlattices and morphology of gold nanocubes are mostly well preserved, when the GNSs are utilized as working electrode with a cyclic potential in the range from −0.8 to 1.0 V vs RHE (all potentials in this work are referenced to the reversible hydrogen electrode (RHE) scale unless noted otherwise) at 50 mV s⁻¹ in 0.1 M $HClO_4$ solution for 20 min (Supplementary Fig. 6). On the contrary, the morphology of CDFs changes significantly during the cyclic voltammetry (CV) testing (Supplementary Fig. 7). GNSs with a narrow gap on the silicon crystal exhibit near-field enhancement due to their coupling plasmonic structures, which provides the possibility for further SEIRAS explorations (Fig. 1d)[22,23].

### Improved performance of GNSs for in situ SEIRAS study of CO₂RR

Rhombic GNSs are further applied as substrates of ATR-SEIRAS for spectroscopic investigations of $CO_2RR$. Potential-dependent SEIRA spectra show that there are several absorption peaks at 3000−3600 cm⁻¹ and 1640 cm⁻¹ in Ar-saturated 0.1 M $HClO_4$, which are attributed to vibrational water stretching and bending modes, respectively (Supplementary Fig. 8)[11,24–26]. All the vibrational peaks shift to lower frequencies with decreasing applied potentials because of the vibrational Stark effect. The peak positions and apparent Stark tuning rates of vibrational water modes in SEIRAS are consistent to those on the surfaces of CDFs[27]. Similar results are observed in neutral electrolytes (Supplementary Fig. 9), which confirms the successful applications of GNSs as substrates for in situ ATR-SEIRAS.

Carbon monoxide is a major intermediate for $CO_2RR$ and the CO coverage is of great importance to produce $C_{2+}$ products via C-C coupling during CORR. The strong adsorption of CO on gold surfaces and relatively intense spectroscopic vibrations make it an ideal probe to investigate the SEIRA effect of different gold films quantitatively. It is noted that similar behaviors of CO adsorption are observed on the surfaces of both CDFs and GNSs in the presence of CO-saturated electrolytes (Fig. 2a and Supplementary Fig. 10). Specifically, the adsorbed CO bands on CDFs shifts from 2140 to 2080 cm⁻¹ with the decreasing potential from 0.9 to −0.3 V (Supplementary Fig. 10a) in 0.1 M $HClO_4$, which agrees well with previous reports[8,28]. Meanwhile, a similar trend is found on GNSs that the adsorbed CO bands shift from 2140 to 2054 cm⁻¹ as the potential decreases from 0.9 to −0.3 V (Supplementary Fig. 10b). The same tendency is observed in neutral electrolytes (Supplementary Fig. 10c, d). We further carried out density functional theory (DFT) calculations to study the adsorption configurations of CO on the surfaces of Au with different facets (Supplementary Fig. 11). The calculated adsorption energy results indicate that CO prefers to adsorb onto the surfaces of Au(110) facets. The simulated IR spectra of CO on the surfaces of Au are in good agreement with the experimental SEIRA spectra. According to previous reports, the Stark shift follows a linear dependence for the sufficiently weak fields (Eq. 1):

$$\triangle v = v - v_0 = -\triangle \mu F \qquad (1)$$

where $v$ ($v_0$) is the frequency of a molecular vibrational mode in the presence (absence) of the filed $F$, and $\Delta \mu$ is the difference in dipole moments (also known as Stark tuning rate) for the molecule in the ground and excited vibrational states[29,30]. It is worth pointing out that besides external fields such as interfacial electric fields produced at the electrochemical surfaces, other local fields such as the solvation field

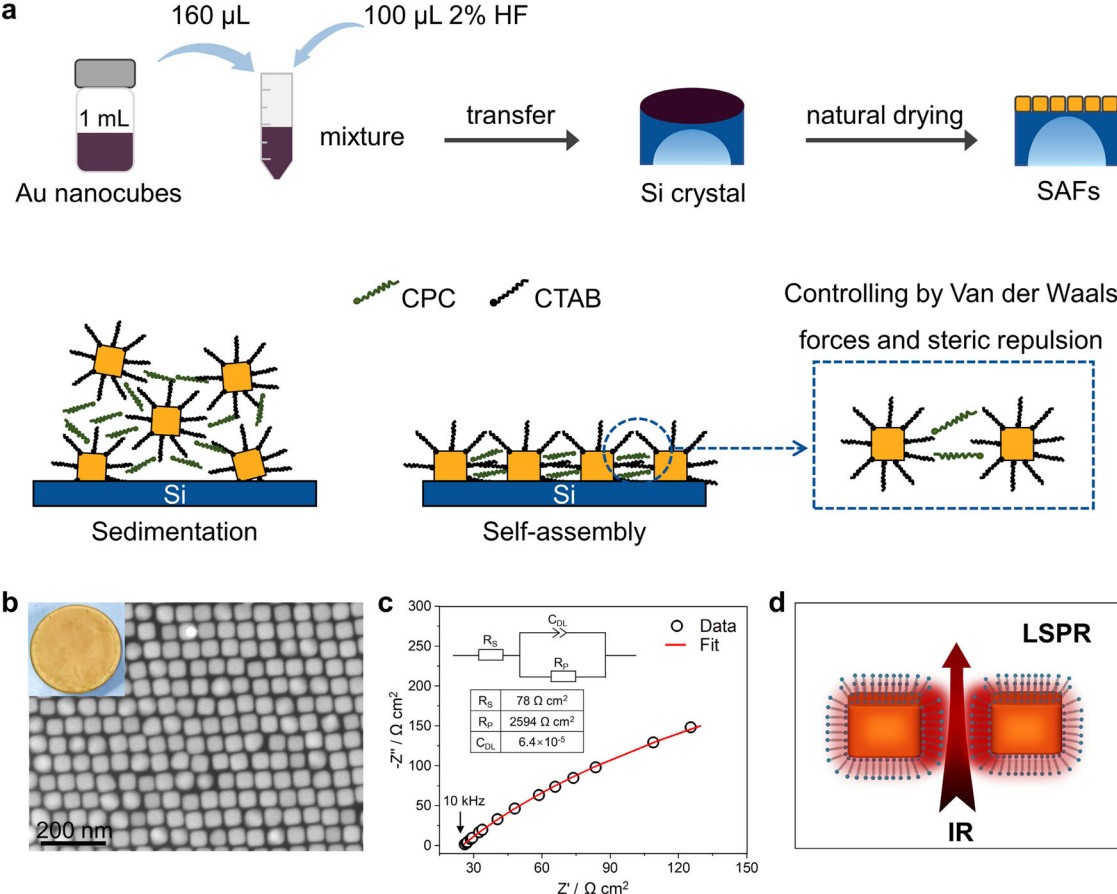

**Fig. 1 | The preparation of rhombic gold nanocube superlattices. a** Schematic illustration depicting the major steps involved in the formation of SAFs, sedimentation, and self-assembly. Highly ordered superlattice films are fabricated by manipulating the Van der Waals force and steric repulsion between Au nanocubes. **b** A typical scanning electron microscopic image of GNSs. The inset present a photo of GNSs with a diameter of 2 cm. **c** Electrochemical impedance spectroscopy of GNSs. **d** Schematic illustration showing near-field enhancement on GNSs.

that originates from solute/solvent interactions also contribute to the vibrational shift[29,31,32]. The Stark shift deviating from linearity (Fig. 2b) could be due to the strong interactions between water molecules and surfactants on the surfaces of GNSs including CPC and CTAB. The apparent Stark tuning rates of adsorbed CO bands on CDFs and GNSs in 0.1 M HClO₄ are determined to be 61 and 63 $cm^{-1}$ $V^{-1}$, respectively (Fig. 2b). The similar peak positions and apparent Stark tuning rates of adsorbed CO bands on CDFs and GNSs further confirm the successful applications of GNSs as alternative substrates for in situ ATR-SEIRAS.

Typically, the enhancement factor is calculated by normalizing the enhanced vibrational signal strength S to the respective nonenhanced vibration signal strength and the number of molecules actively contributing to the signal[13,33]. Here, we introduce a new relative enhancement factor (REF), which is defined as ratio of enhanced vibrational signal strength S on SAFs to that on CDFs (Eq. 2).

$$REF = \frac{S_{SAFs} \times ECSA_{CDFs}}{S_{CDFs} \times ECSA_{SAFs}} \tag{2}$$

In which S represents the maximum peak area of CO adsorption. To eliminate the interference from the number of adsorbed CO, the enhanced vibrational signal strength S needs to be normalized to the electrochemically active surface area (ECSA), on the assumption that the density of adsorbed CO remains the same on different gold films under the same reaction conditions. The ECSAs of CDFs and GNSs are determined to be 0.50 ± 0.21 and 0.41 ± 0.04 cm², based on the reduction peaks of AuO peaks with a double layer correction and a

charge density of 390 μC cm⁻² in 0.1 M HClO₄, respectively (Supplementary Fig. 12)[34–36]. SEIRA spectra collected in 0.1 M HClO₄ show that the peak area of adsorbed CO bands on GNSs are significantly stronger than those on CDFs (Fig. 2c, the peak areas of CO adsorptions on CDFs and GNSs in the potential range from 0.9 to −0.3 V are summarized in Supplementary Table 1). It is worth noting that the vibrational signals on the surfaces of GNSs in ATR-SEIRAS show an enhancement of 6.4 ± 1.7 times compared to those on the CDFs in 0.1 M HClO₄ electrolyte. Similarly, the vibrational signal of adsorbed CO is enhanced by 8.1 ± 1.7 times on the surfaces of GNSs compared to that on the CDFs in 0.5 M KHCO₃ (Supplementary Figs. 10 and 12). The pattern of SAFs are manipulated by controlling the concentration of CPC during the self-assembly process[37,38]. (Supplementary Fig. 13). With the increasing CPC concentration from 1 mM to 5 mM, as-prepared SAFs change from randomly ordered pattern to rhombic superlattices and the gap between two adjacent nanocubes gradually increases from 1 nm to 8 nm. The SAFs become less conductive at a CPC concentration of 10 mM with a resistance of ~350 Ω (Supplementary Fig. 14), which makes them unsuitable for spectroscopic study of CO₂RR. The significantly increased resistance of SAFs is likely due to the increasing distance between two adjacent nanocubes. It is noticed that the adsorbed CO bands normalized to ECSA becomes stronger when the SAFs become more ordered with the increasing CPC concentration to 5 mM (Fig. 2d and Supplementary Fig. 15). The GNSs with a rhombic pattern and a gap of 5−8 nm shows the strongest vibrational signal strength, indicating a close relation between enhancement effect and the randomness of SAFs.

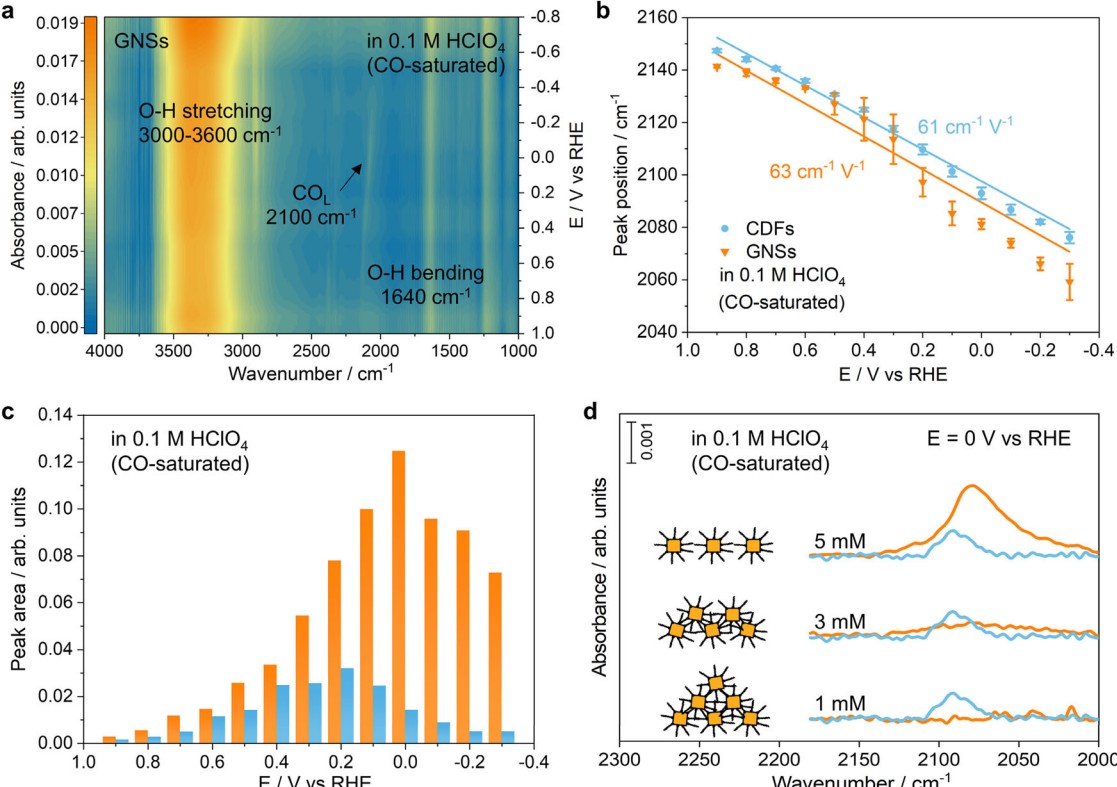

**Fig. 2 | SEIRA spectra on different films for CO adsorption. a** Potential-dependent SEIRA spectra on GNSs in CO-saturated 0.1 M HClO₄ solution. **b** Apparent Stark tuning rates of adsorbed CO bands on CDFs and GNSs. **c** Potential-dependent peak area of adsorbed CO bands on CDFs and GNSs from 0.9 to −0.3 V in 0.1 M HClO₄ (the concentration of CPC for the fabrication of GNSs is 5 mM). **d** The SEIRA spectra showing adsorbed CO bands on CDFs and SAFs fabricated in the presence of different concentrations of CPC. The color scheme in **b** applies to **c** and **d**.

## Enhancement mechanism of SAFs

To reveal the enhancement mechanisms of SAFs for SEIRAS, FDTD simulations are performed on the arrays of Au nanocubes with different randomness (Fig. 3). Noticeably, the electromagnetic field distribution near the Au nanocubes are highly dependent on the randomness of the arrays. The Au nanocubes are randomly arranged and stacked at a CPC concentration of 1 mM (Fig. 3a–c), leading to the particularly weak electromagnetic field strength within the gaps of Au nanocubes. The electromagnetic effect is mostly generated around the edges of the Au nanocube arrays. With the increasing CPC concentration to 3 mM (Fig. 3d–f), the SAFs become more ordered and thus the electromagnetic field strength around the edges of the Au nanocube arrays become stronger. However, the electromagnetic effect between Au nanocubes is still weak due to their random arrangement and stacking. When the CPC concentration further increases to 5 mM (Fig. 3g–i), the SAFs transform into rhombic GNSs, which are arranged in a highly ordered pattern. The electromagnetic field strength becomes much stronger within the gaps and near the edges of Au nanocubes. Meanwhile, the surface enhancement effect reaches maximum. FDTD results show that electromagnetic effect is sensitive to the randomness of Au nanocube arrays, specifically, the rhombic GNSs show the strongest electromagnetic enhancement effect, which is consistent to the ATR-SEIRAS measurements (Fig. 2). The surface enhancement effect ensures their superior plasma-induced absorption capacity, which is the reason why the GNSs at a CPC concentration of 5 mM show the highest plasma-enhanced activity. The interactions between Au nanocubes are greatly enhanced due to the high electromagnetic field strength within the gap, which further enhance the spectroscopic vibrations by activating the adsorbed molecules with direct energy transfer under intense electric fields (Supplementary Figs. 16–22). The improved electromagnetic effect

leads to the significant enhancement of adsorbed CO bands on the GNSs in the SEIRAS. ATR-SEIRAS experiments and FDTD simulations are conducted to study the role of CPC on the surface reaction. There is no peak in the range from 2000 to 2200 cm⁻¹ in Ar-saturated 0.1 M HClO₄ solution and 0.5 M KHCO₃ (Supplementary Fig. 23a, c), indicating that CPC molecule is stable in the potential range from +1.0 to −0.8 V. Afterwards, CO gas was purged into the electrolytes for 30 min. SEIRA spectra show that there is no peak in the range from 2000 to 2200 cm⁻¹ in CO-saturated 0.1 M HClO₄ solution and 0.5 M KHCO₃, suggesting that CPC prevents the adsorption of CO onto the surfaces of gold films (Supplementary Fig. 23b, d). FDTD simulations show that the electromagnetic field strength of GNSs changes little after the modification of CPC, indicating that the contribution of CPC on the SEIRA effect of GNSs is negligible (Supplementary Fig. 24).

## Improved reproducibility of GNSs for in situ SEIRAS study of CO₂RR

Three different GNSs are prepared via the self-assembly process, which are named GNS-1, GNS-2, and GNS-3. SEIRA spectra show that the behaviors of CO adsorption in terms of peak position and intensity are similar to each other on the surfaces of three GNSs (Supplementary Fig. 25). The ECSAs on GNS-1, GNS-2, and GNS-3 are determined to be 0.34, 0.45, and 0.36 cm², respectively. The maximum peak areas of CO adsorptions, ECSAs, and REFs of GNS-1, GNS-2, and GNS-3 are summarized in Supplementary Table 2. The relative standard deviations (RSDs) of the maximum peak areas of CO adsorptions, ECSAs, and REFs of GNS-1, GNS-2, and GNS-3 are determined to be 11.57%, 15.29%, and 4.02%, respectively. We have also fabricated three different CDFs via the wet-chemical deposition approach, which are named as CDF-1, CDF-2, and CDF-3, respectively. SEIRA spectra show that the behaviors of CO adsorption vary significantly in terms of the peak intensity on the

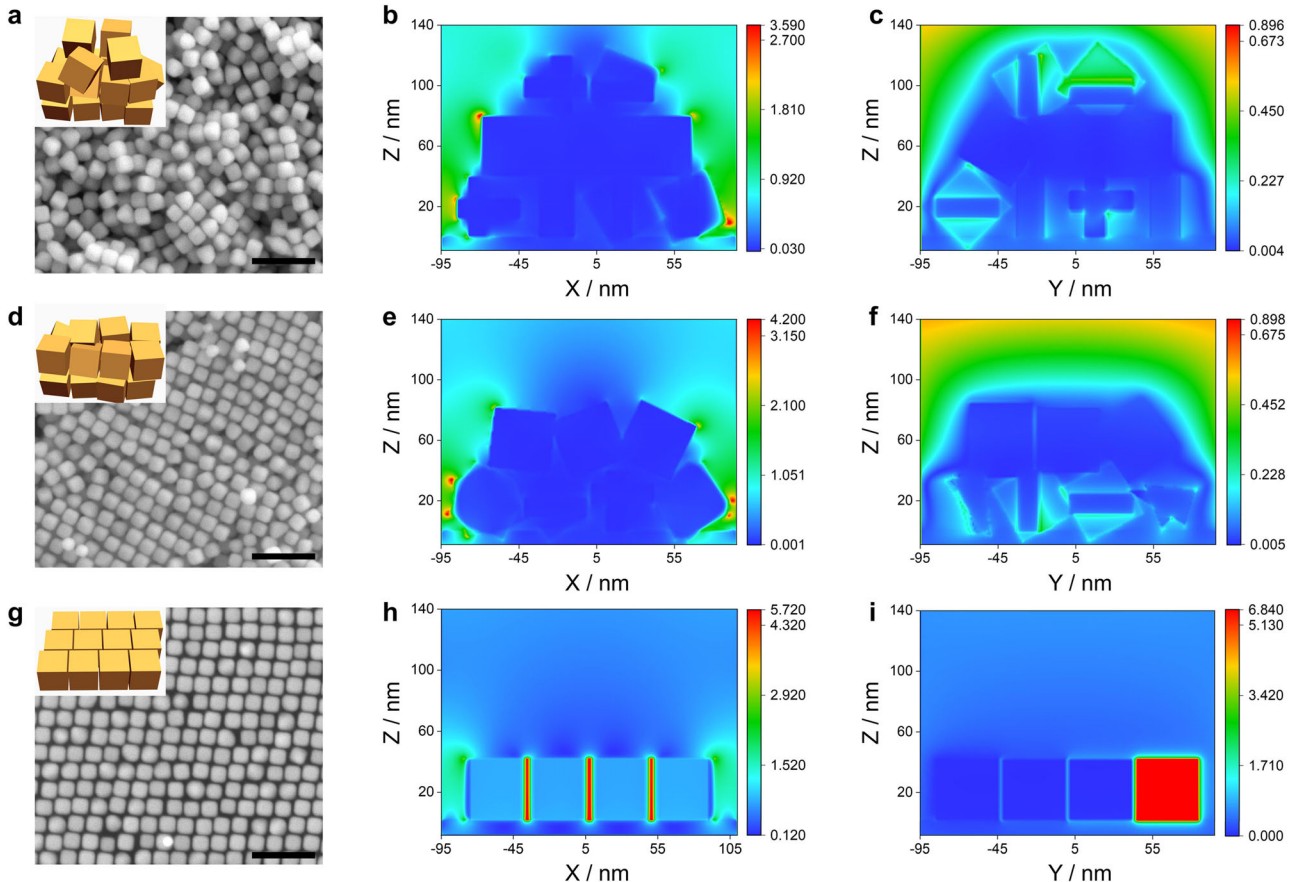

**Fig. 3 | FDTD simulations of the local electromagnetic field for SAFs.** SAFs with different randomness in the presence of 1 mM (**a–c**), 3 mM (**d–f**), and 5 mM CPC (**g–i**). **a, d, g** SEM images of SAFs fabricated in the presence of different concentrations of CPC. Scale bar, 200 nm. The insets in **a**, **d**, and **g** show the schemes of SAFs with different randomness. Simulated near-field enhancement $|E|^2$ of SAFs with different randomness on XZ (**b, e, h**) and YZ (**c, f, i**) planes.

surfaces of three CDFs (Supplementary Fig. 26). The ECSAs on CDF-1, CDF-2, and CDF-3 are determined to be 0.93, 0.86, and 0.82 cm², respectively. The maximum peak areas of CO adsorptions, ECSAs, and REFs of CDF-1, CDF-2, and CDF-3 are summarized in Supplementary Table 3. The RSDs of the maximum peak areas of CO adsorptions, ECSAs, and REFs of CDF-1, CDF-2, and CDF-3 are determined to be 59.19%, 6.40%, and 67.45%, respectively, which are much higher than those on GNSs. Therefore, GNSs prepared via the self-assembly process show significantly improved reproducibility compared to traditional CDFs.

**Improved performance of GNSs for in situ SEIRAS study of CORR**
Cu-based materials with appreciable selectivity for hydrocarbons and oxygenates in CORR have recently attracted much research interest. It has been proposed that the Cu⁺ sites generated from residual oxygen are the active sites for C-C coupling in the CORR. Therefore, the adsorption of CO on copper(I) oxide ($Cu_2O$) is investigated with in situ ATR-SEIRAS using GNSs as substrates. Lead underpotential deposition (Pb UPD) is carried out in 0.1 mM $Pb(ClO_4)_2$ (0.1 M $HClO_4$) to evaluate the ECSA of the $Cu_2O$ catalysts with a charge density of 262 µC cm⁻² and thus exclude the contributions from the ECSA of $Cu_2O$ on the enhanced vibration signals[39,40]. The reduction peak located at −0.2 V vs RHE on the surfaces of both CDFs and GNSs is corresponding to underpotential deposition of Pb²⁺ on the $Cu_2O$ catalysts (Fig. 4a)[41]. The ECSAs for the $Cu_2O$ catalysts deposited on CDFs and GNSs are determined to be 0.59 and 0.68 cm², respectively.

In situ ATR-SEIRAS results indicate that the adsorbed CO bands on the surfaces of CDFs supported $Cu_2O$ catalysts show up at −0.3 V and locates at ~2105 cm⁻¹ (Fig. 4b), which are consistent to the results in previous reports[42,43]. The peak position gradually shifts to 2078 cm⁻¹

when the potential decreases to −1.0 V[44,45]. It is noted that there are multiple distinct CO adsorption sites located in the region of 2000−2105 cm⁻¹, which are attributed to linearly bonded CO[46]. In particular, the CO adsorption bands could been assigned to CO bound to step (~2089 cm⁻¹, high wavenumber), terrace sites (~2073 cm⁻¹, main component), and Cu(100) facets (~2058 cm⁻¹, low wavenumber), respectively. Meanwhile, the adsorbed CO bands appear at the identical position and potential on the surfaces of GNSs supported $Cu_2O$ catalysts (Fig. 4c). The calculated adsorption energy results indicate that CO prefers to adsorb onto the surfaces of Cu(100) and $Cu_2O$(111) facets (Supplementary Fig. 27 and 28). The simulated IR spectra of CO adsorptions on the surfaces of Cu and $Cu_2O$ are in good agreement with the experimental SEIRA spectra. Noticeably, the vibrational intensity of adsorbed CO bands on the $Cu_2O$ catalysts using GNSs is significantly enhanced compared to that using CDFs as the substrates. The REF of GNSs to CDFs is determined to be 2.4 ± 0.5 in 0.1 M $HClO_4$ using CO as the probe molecule for the spectroscopic investigations on the surfaces of $Cu_2O$ catalysts. SEIRAS results in 0.5 M $KHCO_3$ show similar trends to that in the acidic environment. The ECSAs for the $Cu_2O$ catalysts deposited on CDFs and GNSs are determined to be 2.22 and 0.98 cm² in 0.5 M $KHCO_3$, respectively (Fig. 4d). The vibrational intensity of adsorbed CO bands on the GNSs has been enhanced by 18.0 ± 1.3 times compared to that using the CDFs as the substrates in the neutral electrolyte (Fig. 4e, f).

**Spectroscopic investigations of CORR on $Cu_2O$ surfaces in the presence of alkyl iodide**
Alkyl groups are critical intermediates for hydrocarbons and oxygenates in the CORR, however, the detection of which remains

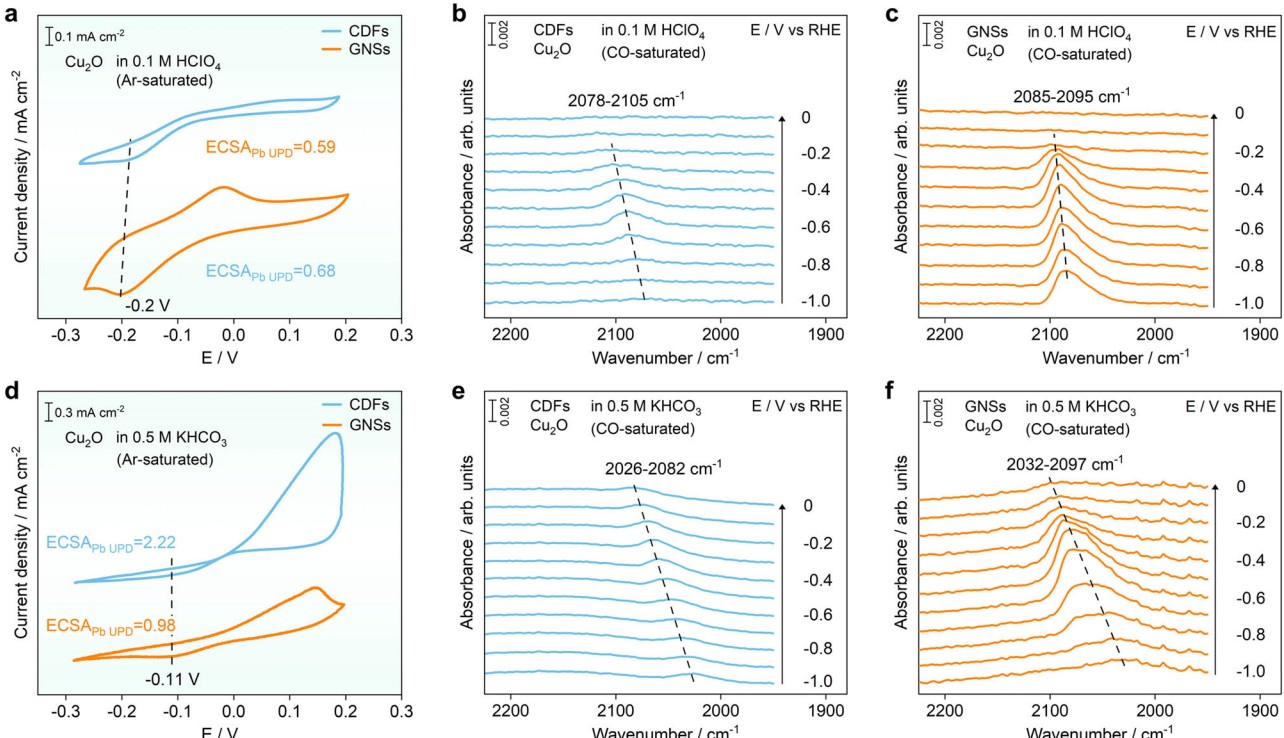

**Fig. 4 | Enhancement effect of adsorbed CO on the surfaces of different substrates supported Cu₂O.** Cyclic voltammograms and in situ ATR-SEIRA spectra of Cu₂O on CDFs (blue lines) and GNSs (orange lines) in different electrolytes: **a–c** 0.1 M HClO₄, **d–f** 0.5 M KHCO₃. The scanning rate for CV measurements is 50 mV s⁻¹.

challenging with SEIRAS due to their inactive spectroscopic responses[47,48]. SEIRA spectra on traditional CDFs show that there are no peaks corresponding to the C-H stretching and CH₂ twisting modes in the Ar-saturated 0.1 M KOH solution before and after the introduction of 0.5 M CH₃I (Fig. 5a and Supplementary Fig. 29a). On the contrary, there are two peaks located at 2980 and 2959 cm⁻¹ on GNSs after introducing CH₃I into the electrolyte, corresponding to C-H stretching of adsorbed CH₃ from CH₃I on Cu₂O surfaces (Fig. 5a)[49–51]. Another two bands located at 1263 and 1241 cm⁻¹ on GNSs are associated with symmetric and antisymmetric CH₂ twisting modes, respectively[51]. It is noted that the intensity of the vibrational peaks corresponding to C-H stretching and CH₂ twisting modes on GNSs decreases with the decreasing potentials. The SEIRA spectra in the presence of CD₃I show that there are two peaks located at 2164 and 2143 cm⁻¹, which are attributed to C-D vibrations on the surfaces of GNSs supported Cu₂O (Supplementary Fig. 30)[52,53]. Reactivity results show that significant amounts of methane and ethane are produced after the introduction of CH₃I into Ar-saturated 0.1 M KOH in the potential range of −0.2 to −0.7 V (Supplementary Fig. 29b). The total Faradaic efficiencies of methane and ethane increase with decreasing potentials, indicating that adsorbed CH₃ and CH₂ from dissociated CH₃I on Cu₂O surfaces are likely the reaction intermediates for the production of methane and ethane through hydrogenation.

With the significantly improved sensitivity, GNSs are further utilized as substrates for in situ spectroscopic study of C-C coupling of the CORR on Cu₂O catalysts in the presence of CH₃I. Potential-dependent SEIRA spectra show that the bands located at 2970, 2940, 1265, and 1242 cm⁻¹, corresponding to C-H stretching and CH₂ twisting modes from adsorbed CH₃, appear on Cu₂O catalysts using GNSs but not CDFs substrates (Fig. 5b and Supplementary Fig. 31)[49–51,54]. There is a band at ~2074 cm⁻¹ on the CDFs and GNSs, which is attributed to linearly bonded CO (CO_L) on the Cu sites of Cu₂O catalysts (Fig. 5c and Supplementary Fig. 32)[55–57]. SEIRA spectra in the presence of ¹³CO-

saturated 0.1 M KOH show that there is a peak located at ~2000 cm⁻¹, which further confirms the linearly bonded ¹³CO on the surfaces of GNSs supported Cu₂O (Supplementary Fig. 33)[45,58]. The weak CO absorption on Cu₂O in the presence of CH₃I is likely due to partially displacement of adsorbed CO by the adsorbed H[7]. Meanwhile, the band located at ~1650 cm⁻¹ changes from symmetric on CDFs to asymmetric on GNSs, which can be deconvoluted into two distinct components via the Gaussian fitting (Fig. 5d). The main component located at 1645 cm⁻¹ is associated with O-H bending. The low frequency component shifts from 1550 to 1581 cm⁻¹ with the applied potential decreasing from −0.2 to −0.7 V, which is attributed to *OCCOH[59–61]. The peak position of *OCCOH vibration shifts to higher frequency with the decreasing potential, which is likely due to the coverage effect. The presence of *OCCOH vibration on the GNSs but not traditional CDFs further confirms the significantly improved sensitivity due to the electromagnetic effect.

Isotopic labeling experiments are performed to reveal the carbon sources of products and the reaction pathways. SEIRAS results show that the peak positions of C-H stretching and CH₂ twisting modes in the presence of ¹³CO-saturated 0.1 M KOH (Supplementary Fig. 34) are almost identical to those in Ar-saturated 0.1 M KOH (Fig. 5a), which further confirms the origination of those bands from CH₃I[58]. The red-shift of the *OCCOH band from 1581 cm⁻¹ in ¹²CO to 1537 cm⁻¹ in ¹³CO suggests that CO participates in the formation of *OCCOH, which is an important intermediate to ethanol via the CORR (Fig. 5e)[62]. The hypothesis is further supported by the NMR analysis of the electrolytes after the CORR in the presence of ¹³CO with CH₃I. Both the ¹H and ¹³C NMR spectra clearly show that there are several distinct peaks corresponding to the ¹²CH₃ group and ¹³CH₂ group in ethanol (Fig. 5f). The peak with a chemical shift of ~1.05 ppm in the ¹H NMR spectra is attributed to ¹²CH₃ from CH₃I[47]. Meanwhile, the peaks at around 3.53 and 3.73 ppm are attributed to ¹³CH₂, which is likely from the hydrogenation of ¹³CO via the CORR. ¹³C NMR spectra show that there are

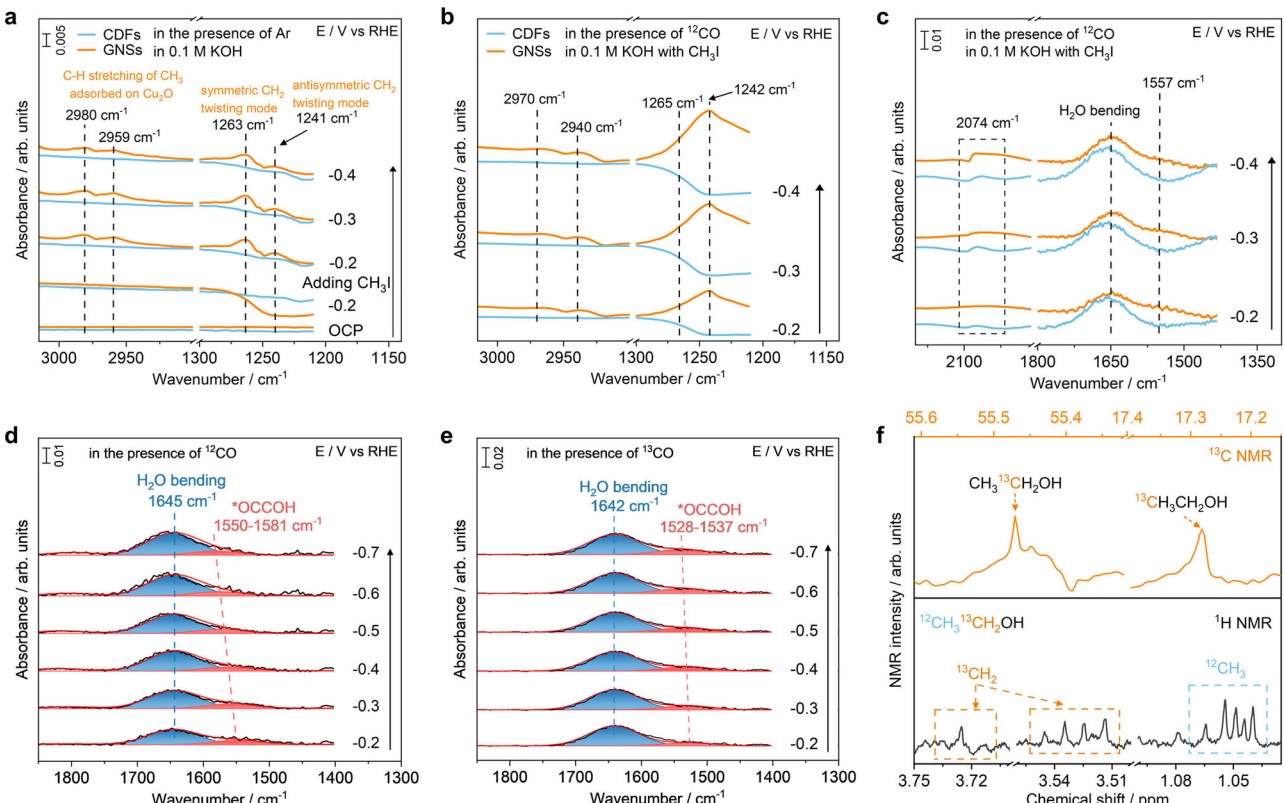

**Fig. 5 | Potential-dependent SEIRA spectra on the surfaces of different substrates supported Cu$_2$O in 0.1 M KOH.** The C-H stretching and CH$_2$ twisting modes from CH$_3$I adsorbed on Cu$_2$O surfaces in the presence of Ar (**a**) and $^{12}$CO (**b**). **c** The adsorbed CO, O-H bending, and *OCCOH vibrations in the presence of $^{12}$CO. The color schemes in **a** and **b** apply to **c**. SEIRA spectra of O-H bending and *OCCOH vibrations on the surfaces of GNSs supported Cu$_2$O catalysts in **d** $^{12}$CO- and **e** $^{13}$CO-saturated 0.1 M KOH with CH$_3$I. **f** $^{1}$H NMR and $^{13}$C NMR peaks corresponding to ethanol produced from the $^{13}$CORR.

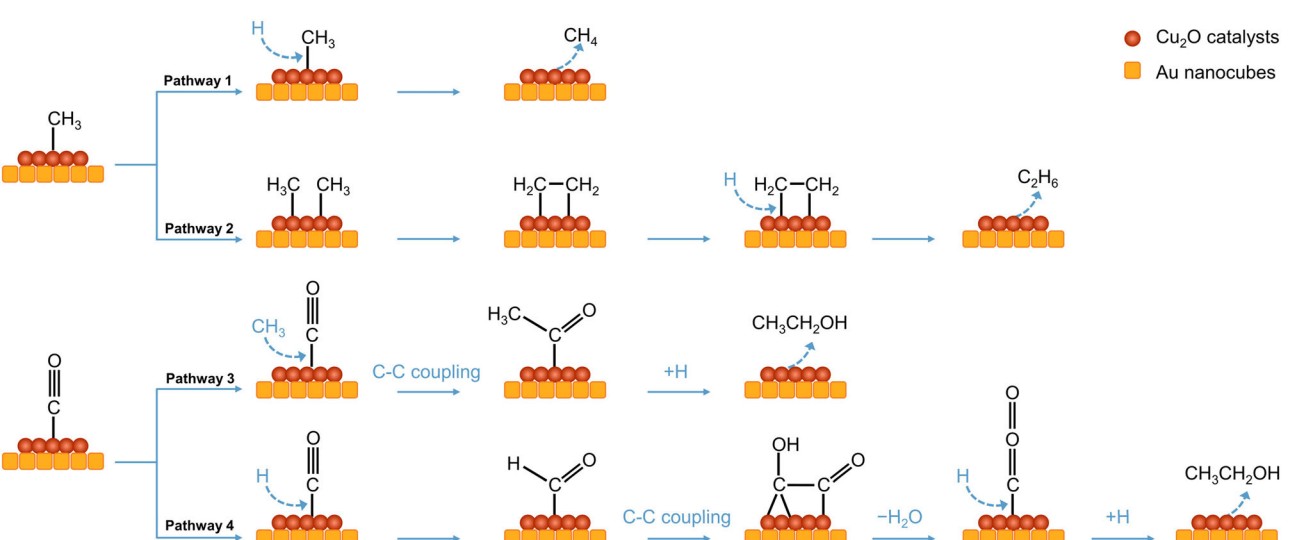

**Fig. 6 | Scheme of reaction pathways for the CORR on Cu$_2$O catalysts in 0.1 M KOH with CH$_3$I electrolyte.** Pathway 1 represents the generation of methane via a hydrogeneration process. Pathway 2 shows the production of ethane from the coupling between two adsorbed CH$_3$. Pathway 3 represents the coupling of CH$_3$ group with adsorbed CO molecule to produce ethanol. Pathway 4 shows the coupling of *CHO and adsorbed CO molecule for the generation of ethanol.

two peaks at around 17.28 and 55.47 ppm, corresponding to the $^{12}$CH$_3$ and $^{13}$CH$_2$ in ethanol, respectively, which confirms that ethanol is produced via the coupling of *$^{13}$CO and *$^{12}$CH$_3$ on the Cu$_2$O catalysts.

Combined with spectroscopic and isotopic labeling experiments, the possible reaction pathways in Ar/CO-saturated 0.1 M KOH with CH$_3$I solution on Cu$_2$O surfaces are speculated (Fig. 6). Methane is produced from adsorbed CH$_3$ on Cu$_2$O surfaces through a hydro-generation process (Pathway 1). Meanwhile, ethane is generated from the coupling between two adsorbed CH$_3$ on Cu$_2$O surfaces (Pathway 2). In the presence of CO, the adsorbed CO molecule is converted into *COCH$_3$ via the coupling with CH$_3$ group, and then the produced *COCH$_3$ is hydrogenated to form ethanol (Pathway 3). This is

consistent with results from recent Raman studies by Xu and co-workers[58]. In the CORR, adsorbed CO is hydrogenated into *CHO and then form a hydrogenated CO dimer intermediate (*OCCOH) by coupling with another adsorbed CO. The intermediate *OCCOH is then converted into ethanol via a series of dehydration and hydrogenation steps (Pathway 4).

## Discussion

We have demonstrated a novel approach to fabricating gold superlattice films through self-assembly of gold nanocubes for in situ ATR-SEIRAS study on CORR, with significantly enhanced SEIRA effect of around one order of magnitude. The SEIRA effect of SAFs is highly dependent on the randomness of the Au nanocube arrays, which increases drastically with the decreasing randomness. The superlattice with an interspace of 5−8 nm shows the highest SEIRA effect with REFs of $6.4 \pm 1.7$ and $8.1 \pm 1.7$ in 0.1 M $HClO_4$ and 0.5 M $KHCO_3$, respectively. FDTD simulations reveal that the electromagnetic effect accounts for the significantly improved spectroscopic vibrations on the SAFs. With $Cu_2O$ as the catalysts for CORR, the vibrations of adsorbed CO are significantly enhanced by $2.4 \pm 0.5$ and $18.0 \pm 1.3$ times as compared to those on traditional CDFs in the acidic and neutral electrolytes, respectively. Moreover, the stretching and twisting vibrations of alkyl groups are detected on $Cu_2O$ catalysts using GNSs but not CDFs as the substrates. Combined with isotopic labeling experiments, SEIRAS results on the GNSs reveal that the coupling reactions involving abundant surface species such as adsorbed CO and $CH_3$ group from $CH_3I$ yield detectable amounts of ethanol via the CORR. Taken together, this work provides an attractive strategy for revealing the reaction mechanisms of surface-mediated electrochemical reactions with high sensitivity and reproducibility.

## Methods

### Chemicals and materials

Gold chloride trihydrate ($HAuCl_4·2H_2O$) was purchased from Shanghai Chemical Reagent. Hydrofluoric acid (HF), perchloric acid ($HClO_4$), sodium sulfite ($Na_2SO_3$, 99%), sodium thiosulfate ($Na_2SO_4$, 99.99%), ammonium chloride ($NH_4Cl$, 99.5%), ammonium fluoride ($NH_4F$, 98%), sodium hydroxide (NaOH, 97%), potassium bicarbonate ($KHCO_3$, 99.7%) and dimethyl sulfoxide (DMSO, 99.5%) were purchased from Shanghai Aladdin Biochemical Technology Co., Ltd. Hexadecyl-trimethylammonium bromide (CTAB, 99%), sodium borohydride ($NaBH_4$, 98%), L-ascorbic acid (AA, 99.7%), and potassium hydroxide (KOH, 96%) were purchased from Sinopharm Chemical Reagent Co., Ltd. Cetylpyridinium chloride (CPC, 98%) was purchased from Sangon Biotech (Shanghai) Co., Ltd. Copper (I) oxide ($Cu_2O$, 99%) was purchased from Bidepharm (Shanghai). Lead(II) perchlorate trihydrate ($Pb(ClO_4)_2·2H_2O$, 97%) and deuterium oxide ($D_2O$, 99.9%) were purchased from Shanghai Macklin Biochemical Co., Ltd. Methylene iodide ($CH_3I$, 99.5%) was purchased from Energy Chemical. Carbon Dioxide ($CO_2$, 99.999%), carbon monoxide (CO, 99.999%), and argon (Ar, 99.999%) were purchased from Wuhan Zhongxin Ruiyuan gas Co., Ltd. Nafion-117 membrane was purchased from Gaoss Union. Double distilled water ($DI-H_2O$) was obtained using an automatic pure water distiller (SZ-96, Shanghai Xiande Experimental Instrument Co., Ltd.).

### Characterizations

Scanning electron microscopy images were captured on a field emission electron microscope (Gemini SEM 300, Carl Zeiss). UV-vis absorption spectra were recorded on Shimadzu UV-2600i. Electrochemical experiments were carried out using a Princeton VersaSTAT 3 F potentiostat (Princeton Applied Research). All spectroscopic measurements were collected with 4 $cm^{-1}$ resolution and at least 128 coadded scans using a FTIR spectrometer (Nicolet iS50, Thermo Scientific) equipped with a liquid nitrogen-cooled MCT detector. Nuclear magnetic resonance spectra were collected on the Nuclear Magnetic

Resonance Spectrometer (Bruker Avance NEO 600 MHz). FDTD calculations were performed using commercial software (FDTD solutions, Lumerical Solutions, Inc.).

### Preparation of CDFs

CDFs were prepared using an electroless chemical plating method on an undoped Si prism by following the previously reported procedure[61]. The prism was first polished using 0.05 μm $Al_2O_3$ and sonicated in acetone and water to remove residue. Following cleaning, the reflecting plane of the prism was immersed in $NH_4F$ (40%) for 120 s to remove the oxide layer and create a hydrogen-terminated surface to improve adhesion of the gold film. The Si surface was then immersed in a 4.4:1 by volume mixture of 2% HF and Au plating solution consisting of 5.75 mM $NaAuCl_4·2H_2O$, 0.025 M $NH_4Cl$, 0.075 M $Na_2SO_3$, 0.025 M $Na_2S_2O_3·5H_2O$, and 0.026 M NaOH at 55 °C for 5 min. SEM images show that the thickness of CDFs is ~52 nm (Supplementary Fig. 2).

### Synthesis of gold nanocubes

Gold nanocubes were synthesized using the seed-mediated growth method[19]. Step 1: 0.125 mL of 10 mM $HAuCl_4·3H_2O$ solution was added into 3.75 mL of 100 mM CTAB solution under vigorous stirring. Subsequently, 0.3 mL of freshly prepared ice-cold $NaBH_4$ solution (10 mM) was added into the mixture solution within 3 min and then kept at 30 °C water for 2 h, which was diluted by 250 times with $DI-H_2O$ and then used as seed solution for the synthesis of gold nanocubes. Step 2: 16 mL of 100 mM CTAB and 2 mL of 10 mM $HAuCl_4·3H_2O$ were added into 100 mL $DI-H_2O$ under stirring. Then, 9.5 mL of 100 mM AA and 100 μL seed solution were subsequently added into the mixture solution and kept at 30 °C for 12 h to allow for the growth of Au nanocubes. After washed three times with $DI-H_2O$, as-synthesized gold nanocubes were then dispersed in 2 mL 1 mM, 3 mM, or 5 mM CPC solutions, respectively, for further self-assembly process.

### Preparation of self-assembled gold films (SAFs)

As-synthesized gold nanocubes were used as building blocks without further purification. The solution of Au nanocubes was mixed with 2 wt % HF solution at a volume ratio of 1.6:1 to form the plating solution. The plating solution was then drop-casted onto the surfaces of Si prisms and left in an undisturbed open-air environment for the formation of SAFs[19].

### Electrochemical experiments

All electrochemical experiments were conducted in a customized spectroelectrochemical cell[63]. CDFs and SAFs on the Si prisms were used as the working electrode. Graphite rod and Ag/AgCl (3.0 M KCl, Gaoss Union) electrodes were used as counter electrode and reference electrode for all electrochemical experiments, respectively. The working electrode is separated from the counter electrode using the Nafion-117 membrane.

### In situ ATR-SEIRAS experiments

In situ ATR-SEIRAS experiments were conducted in a customized spectroelectrochemical cell[63]. To investigate the $CO_2RR$ on Au films, Ar gas was constantly purged into the electrolytes for 30 min to remove any dissolved $CO_2$. The reference spectra were collected at 1.0 V vs RHE in Ar-saturated electrolytes. For spectroscopic study of the CORR on the surfaces of $Cu_2O$, 1 mg of $Cu_2O$ catalysts were dispersed in a mixture of 0.8 mL $DI-H_2O$, 0.2 mL of isopropanol, and 10 μL of 5% Nafion under ultrasonication for 30 min to produce an ink solution. The suspension was then placed on the CDFs and SAFs for further SEIRAS studies. Ar gas was constantly purged into the electrolyte for 30 min to remove any dissolved $CO_2$. Reference spectra were collected at OCP in Ar-saturated 0.1 M KOH. Then, 0.5 M $CH_3I$ was introduced into the electrolyte at −0.2 V and SEIRA spectra were collected in the potential

range of −0.2 to −0.7 V. Afterwards, CO gas was constantly purged into the cell for the investigation of CORR.

## FDTD simulations

The enhancement mechanism of SAFs was explored by FDTD simulation. To ensure open boundaries in the simulated nanocubes, the FDTD method under perfectly matched layers (PML) boundaries were used. FDTD calculations were performed using commercial software (FDTD solutions, Lumerical Solutions, Inc.) to simulate the local electromagnetic field of the cubic Au nanocrystals. The software's material database provided the optical constants of Au-Palik. The standard Palik material was selected in this simulation. The total-field scattered field (TFSF) light source was set downward along the z-axis with a direction of 90 degree to the flat surface of Au nanocubes. The side length of Au nanocubes was fixed at 40 nm. The uniform mesh size of 1 nm (dx, dy, and dz) was chosen to ensure the accuracy of the simulation results.

## Isotopic labeling experiments

$^{13}CO$ was employed in the CORR to verify the intermediates of C-C coupling step combined with in situ SEIRAS. Ar gas was constantly purged for 15 min before electrolysis. Then, $^{13}CO$ gas was purged at 5 mL min$^{-1}$ for 20 min in the sealed spectroelectrochemical cell. SEIRAS experiments was conducted on the rhombic gold nanocube superlattices (GNSs) supported $Cu_2O$ catalysts under applied potentials. The liquid products were analyzed by $^1H$ and $^{13}C$ NMR. The NMR samples were prepared by mixing 500 μL of the electrolyte post electrolysis with 100 μL $D_2O$ containing 50 ppm DMSO as an internal standard.

## Data availability

All data generated or analyzed during this study are included in the published article and its supplementary information files.

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

## Acknowledgements

XiaojuY., L.Z., Z.Y., C.H., Q.Y., and XuanY. acknowledge the support from the National Natural Science Foundation of China (Grant. No. 22204054), HUST Academic Frontier Youth Team grant (Grant No. 2019QYTD11), Knowledge Innovation Program of Wuhan-Shuguang, and Wuhan Talented Youth Program. C.R. and B.Z. acknowledge the National Natural Science Foundation of China (Grant. No. 52105145, No. 12274124). Z.W., Q.Z., and Y.Z. acknowledge the support from the National Natural Science Foundation of China (No. 21974103) and the start-up funds of Wuhan University. Thanks to engineer Wei Xu in Optoelectronic Micro&Nano Fabrication and Characterizing Facility, Wuhan National Laboratory for Optoelectronics of Huazhong University of Science and Technology for the support in device fabrication.

## Author contributions

XuanY. conceived and designed the project. XiaojuY. prepared the gold superlattices and performed the electrochemical/spectroscopic experiments. C.R., F.-Z.X., and B.Z. carried out the theoretical calculations. L.Z. helped with the analysis of spectroscopic data. Z.Y. helped with the preparation of gold superlattices. Z.W., Q.Z., and Y.Z. conducted the $^{13}$C-NMR measurements. C.H. helped with the analysis of NMR results. Q.Y. conducted the TEM measurements. XiaojuY., B.X., B.Z., and XuanY. analyzed the experimental data and prepared the manuscript. All authors reviewed and contributed to the manuscript.

## Competing interests

The authors declare no competing interests.
