## [Peer Review File · Nature Communications]

REVIEWER COMMENTS

Reviewer #1 (Remarks to the Author):

With the highly ordered rhombic gold nanocube superlattices, authors claimed that a significantly enhanced SEIRAS substrate is obtained, which facilitated the reaction mechanism exploration of CORR on Cu-based catalysts. The work more like reports an interesting result, and the discussion related interfacial mechanism needs further improvement.

1. Authors may have different understanding about the difference between “operando” and “in-situ”. To my point, the experiment carried here is not “operando” but “in-situ”.
2. How about the role of surfactant on the surface reaction? Was it considered in FDTD simulation?
3. The higher roughness doesn't mean the stronger IR enhancement. Authors should read more related literatures.
4. The adsorption configuration of CO is little discussed, which is weird for the CO related SEIRAS research.

Reviewer #2 (Remarks to the Author):

In this manuscript, the authors fabricated superlattices of gold nanocubes, and used these structures for enhancement of vibrational spectroscopy features during CORR. The approach is interesting, but several questions remain before this manuscript can be published in Nature Communications in my opinion, as I will detail below.

1. Can the authors rule out the effect of the Au nanocubes on the spectra in the presence of Cu₂O, or on the catalytic performance? Could it be that the reaction mechanism is influenced by the presence of Au? What is the catalytic performance of the reported structures?
2. Homogeneous enhancement would be a strong selling point for the presented superlattices over disordered Au substrates. The authors are encouraged to present homogeneous enhancement through spatial investigations.
3. the definition of REF (equation 1) is not really valid in my opinion, because the intensity of the studied features largely depends on enhancement which is a spatially varying parameter. Without spatial investigations, the REF equation has no real physical meaning in my opinion. It would be better to relate the vibrational strength to some internal standard, like water vibrations or electrolyte species, and then compare the relative enhancement factor.

4. In Figure 2c, the comparison of the area is speculative if the integrated peaks are not shown. What integration area was used? If a feature is weak but very broad on a large background, the area calculation can still give a large value. Were the spectra background corrected? And for which concentration CPC was this data collected (I am assuming 5 mM, but this should be indicated).
5. The changes in the slope of the Stark shift in Figure 2b should be discussed. Why does it deviate from linearity? Why did the authors fit it to a linear fit, while clearly it is not linear? Is the difference of 61 and 63 cm⁻¹V⁻¹ then significant?
6. The potentials in Figure 2a are not indicated in the figure, this should be corrected.
7. Why is no peak observed in Figure 2d for the disordered films with 1 mM CPC? This would almost resemble a CDF right? Why is the peak on CDF observed and not on disordered superlattices?
8. When Cu₂O is reduced to Cu, copper itself will give enhancement as well. Is the analysis in Fig 4 corrected for the enhancement of Cu itself? Did the authors perform similar experiments on pure Cu₂O without Au substrates? Can the authors exclude enhancement from Cu, for example through spatial investigations or blank measurements? Or perhaps Raman measurements to study the Au-C and Cu-C vibrations at low Raman shifts?
9. What is the Stark tuning rate on Cu? Also, the features in Figure 4 seem to consist of multiple bands (at least two), which are not discussed in detail in the manuscript or deconvoluted. Why are the Stark rates different in different electrolytes (compare Fig 4c and 4f).
10. The proposed mechanisms in Figure 5 are a bit speculative in my opinion, because the NMR features ascribed to ethanol are quite weak, as well as the corresponding vibrational features in Fig 5d,e for *OCCOH or *CO (Fig. 5c). Are these features confirmed to be C-containing intermediates through ¹³C labeling experiments? Different peak positions are ascribed to the *OCCOH intermediates in Figure 5d and e, but not properly discussed. How were these features deconvoluted from the strong OH bending nearby? What shift is expected based on theory, and does this match with the assigned peak positions? I encourage the authors to include more discussion on this dataset, and potentially conduct more experiments to also study the labeling effect on *CO and *CH₂/*CH₃.

Overall, I look forward to receiving a revised manuscript that takes the points mentioned above into account.

Reviewer #3 (Remarks to the Author):

This article by Yang et al. reports highly ordered rhombic gold nanocube superlattices (GNSs) as an alternative substrate for operando SEIRAS. It is featured with its high sensitivity, with the vibrations of CO probe molecule being enhanced by 2.4 and 18 times compared to those on traditional chemically deposited Au films in acidic and neutral electrolytes. It is highly recommended to develop such an

interesting and powerful method for addressing important spectroscopic issues in complicated CO₂ electroreduction systems. This manuscript for sure has great merit for this field, and should be published in Nature Communications.

Before acceptance, I would suggest the authors to consider carefully the comments below.

1. As the authors mentioned, in addition to high sensitivity, good reproducibility is also of great importance for a well-recognized experimental method. The SEIRAS experiments using traditional chemically deposited Au films often suffers from bad reproducibility of Au films due to not well controlled preparation procedures. It would be good to show such reproducibility quantitatively in this work.
2. It would be also appreciated if the authors comment its advantages over the shell-isolated nanoparticle-enhanced Raman spectroscopy (SHINERS) technique for detecting species during CO₂ electroreduction.
3. Fig.1c, it is not clear that why the EIS indicates an excellent conductivity. It is good to also consider the electrode area (with a unit of $\Omega\cdot\text{cm}^2$) and compare to a reference/literature value.
4. Supplementary Fig. 9, the CO adsorption band appears at up to 2140 cm^{-1} which is almost that of free CO in gas. Is that convincing? CO adsorption on Au also occurs at up to +0.8 V vs RHE. However, CO adsorption on Au is usually weak.
5. Why is there an obvious shift of CO adsorption bands on CDFs and GNSs at 5 mM?
6. Is there any Cu₂O solution in acid (0.1 M HClO₄ solution)?
7. Why was C-C coupling of the CORR on Cu₂O catalysts studied in the presence of CH₃I? As the authors mentioned (Fig.6, pathway 4), -CH₃ may be not directly involved in the formation of some products such as ethanol. And, it is not convincing that no signals assigned to -CH₃ are observed using traditional chemically deposited Au film substrate, in a solution containing 0.5 M CH₃I. The concentration of CH₃I should be also high enough on and near the electrode surface.

Reviewer #1

Recommendation:

“With the highly ordered rhombic gold nanocube superlattices, authors claimed that a significantly enhanced SEIRAS substrate is obtained, which facilitated the reaction mechanism exploration of CORR on Cu-based catalysts. The work more like reports an interesting result, and the discussion related interfacial mechanism needs further improvement.”

Response: We thank the reviewer for the positive assessment and constructive suggestions of our work.

1) *“Authors may have different understanding about the difference between “operando” and “in-situ”. To my point, the experiment carried here is not “operando” but “in-situ”.*

Response: We thank the reviewer for the valuable comments. Indeed, the experiment carried in this work is “in-situ” but not “operando” and we have changed “operando” to “in-situ” in the revised manuscript.

Action: We have changed “operando” to “in situ” in revised manuscript.

2) *“How about the role of surfactant on the surface reaction? Was it considered in FDTD simulation?”*

Response: We thank the reviewer for the valuable comments. There are several different roles that CPC plays in this work. First, CPC serves as a surfactant to manipulate the randomness and gaps of rhombic gold nanocube superlattices (GNSs) during the self-assembly process (Fig. 1a). Control experiments are conducted to reveal the roles of CPC on the surface reaction. 5 mM CPC solution was utilized to modify the surfaces of chemically deposited gold films (CDFs). Potential-dependent SEIRA spectra on CPC-modified CDFs show that there is no peak in the range from 2000 to 2200 cm^{-1} in Ar-saturated 0.1 M HClO_4 solution and 0.5 M KHCO_3 (Supplementary Fig. 23, a and c), indicating that CPC molecule is stable in the potential range from +1.0 to -0.8 V. Afterwards, CO gas was purged into the electrolytes for 30 min. SEIRA spectra show that there is no peak in the range from 2000 to 2200 cm^{-1} in CO-saturated 0.1 M HClO_4 solution and 0.5 M KHCO_3 , suggesting that CPC prevents the adsorption of CO onto the surfaces of gold films (Supplementary Fig. 23, b and d). FDTD simulations are further conducted to study the SEIRA effect of GNSs before and after the modification of CPC (Supplementary Fig. 24). It is found that electromagnetic field strength of GNSs changes little after the modification of CPC, indicating that the contribution of CPC on the SEIRA effect of GNSs is negligible.

Supplementary Fig. 23 | Potential-dependent SEIRA spectra on the surfaces of CPC-modified CDFs (the concentration of CPC is 5 mM) in different electrolytes. a Ar-saturated 0.1 M HClO₄, **b** CO-saturated 0.1 M HClO₄, **c** Ar-saturated 0.5 M KHCO₃, and **d** CO-saturated 0.5 M KHCO₃.

Supplementary Fig. 24 | FDTD simulations of the local electromagnetic field for GNSs before (a, c, and e) and after (b, d, and f) the modification of CPC. Simulated near-field enhancement $|E|^2$ on XY planes (a and b), XZ planes (c and d), and YZ planes (e and f).

Action: We have attached the SEIRA spectra on the surfaces of CPC-modified CDFs as the Supplementary Fig. 23 on Page 25 in the revised supplementary information. We have also added the following sentences on Page 13 in the revised manuscript when discussing the role of CPC on the surface reaction.

“ATR-SEIRAS experiments and FDTD simulations are conducted to study the role of CPC on the surface reaction. There is no peak in the range from 2000 to 2200 cm^{-1} in Ar-saturated 0.1 M HClO_4 solution and 0.5 M KHCO_3 (Supplementary Fig. 23, a and c), indicating that CPC molecule is stable in the potential range from +1.0 to -0.8 V. Afterwards, CO gas was purged into the electrolytes for 30 min. SEIRA spectra show that there is no peak in the range from 2000 to 2200 cm^{-1} in CO-saturated 0.1 M HClO_4 solution and 0.5 M KHCO_3 , suggesting that CPC prevents the adsorption of CO onto the surfaces of gold films (Supplementary Fig. 23, b and d). FDTD simulations show that the electromagnetic field strength of GNSs changes little after the modification of CPC, indicating that the contribution of CPC on the SEIRA effect of

GNSs is negligible (Supplementary Fig. 24).”

3) “*The higher roughness doesn’t mean the stronger IR enhancement. Authors should read more related literatures.*”

Response: We thank the reviewer for the valuable comments and fully agree to the reviewer’s opinion that the higher roughness doesn’t necessarily translate to the stronger IR enhancement. According to previous reports (Refs: Moskovits, M. *Rev. Mod. Phys.* **1985**, 57, 783–826; Osawa, M. *et al.*, *Surf. Sci. Lett.* **1992**, 262, L118–L122), rough metal island film is necessary to achieve significantly enhanced vibrations of adsorbed species. We are sorry for the confusion and have changed the statement on Page 3 in the revised manuscript.

Action: We have changed our statement “Typically, SEIRAS requires rough metal island films as the substrates to achieve significantly enhanced vibrations of adsorbed species.” to “Typically, films with rough metal islands are necessary to achieve significantly enhanced vibrational signals of adsorbed species.^{9,10}” on Page 3 in the revised manuscript.

4) “*The adsorption configuration of CO is little discussed, which is weird for the CO related SEIRAS research.*”

Response: We thank the reviewer for the valuable comments. The primary focus of the current manuscript is the fabrication of highly ordered rhombic gold nanocube superlattices (GNSs) as substrates for SEIRAS with high sensitivity and reproducibility. Using the GNSs substrates, the reaction mechanisms for C-C coupling of CORR on Cu-based catalysts are revealed. Therefore, the adsorption configuration of CO is little discussed in the manuscript. Experiments and DFT calculations are conducted to study the adsorption configuration of CO on the surfaces of Au, Cu, and Cu₂O. SEIRA spectra show that the CO adsorption on CDFs shifts from 2140 to 2080 cm⁻¹ with the decreasing potential from 0.9 to -0.3 V (Supplementary Fig. 10a) in 0.1 M HClO₄, which agrees well with previous reports. (Ref: Blizanac, B. B. *et al.*, *J. Am. Chem. Soc.* **2004**, 126, 10130–10141). Meanwhile, a similar trend is found on GNSs that the adsorbed CO bands shift from 2140 to 2054 cm⁻¹ as the potential decreases from 0.9 to -0.3 V (Supplementary Fig. 10b). The same tendency is observed in neutral electrolytes (Supplementary Fig. 10, c and d). We further carried out density functional theory (DFT) calculations to study the adsorption configurations of CO on the surfaces of Au, Cu, and Cu₂O with different facets (Supplementary Fig. 11, 28 and 29). The calculated adsorption energy results indicate that CO prefers to adsorb onto the surfaces of Au(110) facets, though the adsorption energies on different facets do not differ significantly. The simulated IR spectra of CO on the surfaces of Au are in good agreement with the experimental SEIRA spectra.

SEIRAS results indicate that the adsorbed CO bands on the surfaces of CDFs supported Cu₂O catalysts shows up at -0.3 V and locates at ~2105 cm⁻¹. The peak position gradually shifts to 2078 cm⁻¹ when the potential decreases to -1.0 V (Fig. 4b), which is consistent to previous reports (Refs: Malkani, A. S. *et al.*, *ACS Catal.* **2019**, 9, 474–478; Wuttig, A. *et al.*, *ACS Cent. Sci.* **2016**, 2, 522–528). Meanwhile, the adsorbed

CO bands appear at the identical position and potential on the surfaces of GNSs supported Cu₂O catalysts (Fig. 4c). The calculated adsorption energy results indicate that CO prefers to adsorb onto the surfaces of Cu(100) and Cu₂O(111) facets. The simulated IR spectra of CO on the surfaces of Cu and Cu₂O are in good agreement with the experimental SEIRA spectra.

Action: We have added the simulated IR spectra of CO adsorptions on the surfaces of Au, Cu, and Cu₂O with different facets as Supplementary Fig. 11, 28 and 29 on Page 12, 32 and 33 in the revised supplementary information. Also, we have added the following sentences when discussing the CO adsorptions on the surfaces of Au, Cu, and Cu₂O on Page 8 and 16 in the revised manuscript.

“We further carried out density functional theory (DFT) calculations to study the adsorption configurations of CO on the surfaces of Au with different facets (Supplementary Fig. 11). The calculated adsorption energy results indicate that CO prefers to adsorb onto the surfaces of Au(110) facets. The simulated IR spectra of CO on the surfaces of Au are in good agreement with the experimental SEIRA spectra.”

“The calculated adsorption energy results indicate that CO prefers to adsorb onto the surfaces of Cu(100) and Cu₂O(111) facets (Supplementary Fig. 28 and 29). The simulated IR spectra of CO on the surfaces of Cu and Cu₂O are in good agreement with the experimental SEIRA spectra.”

Supplementary Fig. 11 | Simulated IR spectra of CO adsorptions on the surfaces of Au with different facets: a Au(100), b Au(110), and c Au(111). The insets in a, b, and c show the adsorption configurations, adsorption energy, and the bond length of CO.

Supplementary Fig. 28 | Simulated IR spectra of CO adsorptions on the surfaces of Cu with different facets: a Cu(100), b Cu(110), and c Cu(111). The insets in a, b, and c show the adsorption configurations, adsorption energy, and the bond length of CO.

Supplementary Fig. 29 | Simulated IR spectra of CO adsorptions on the surfaces of Cu₂O with different facets: a Cu₂O(100), b Cu₂O(110), and c Cu₂O(111). The insets in a, b, and c show the adsorption configurations, adsorption energy, and the bond length of CO.

Reviewer #2

Recommendation:

“In this manuscript, the authors fabricated superlattices of gold nanocubes, and used these structures for enhancement of vibrational spectroscopy features during CORR. The approach is interesting, but several questions remain before this manuscript can be published in Nature Communications in my opinion, as I will detail below.”

Response: We thank the reviewer for the positive assessment and constructive suggestions of our work.

1) “Can the authors rule out the effect of the Au nanocubes on the spectra in the presence of Cu₂O, or on the catalytic performance? Could it be that the reaction mechanism is influenced by the presence of Au? What is the catalytic performance of the reported structures?”

Response: We thank the reviewer for the valuable comments. According to previous reports, the catalytic performance and SEIRA spectra of oxide-derived Cu catalysts would be influenced in the presence of Au (Ref: Malkani, A. *et al.*, *ACS Catal.* **2019**, 9, 474–478). The SEIRAS results in this work show that the behavior of CO adsorptions on CDFs supported Cu₂O catalysts is identical to that on GNSs supported Cu₂O catalysts, suggesting that GNSs show similar effect on the spectra of Cu₂O to that of CDFs.

Control experiments are further conducted to understand whether the presence of Au would affect the catalytic performance of Cu₂O (Fig. R1 in the response letter). The reactivity results in CO-saturated 0.1 M KOH with CH₃I at -0.7 V vs RHE show that the production rates on the surfaces of GNSs are similar to those on CDFs. Meanwhile, the production rates on the surfaces of GNSs supported Cu₂O are similar to

those on CDFs supported Cu_2O , but different from those on carbon paper supported Cu_2O . Therefore, the presence of Au would affect the catalytic performance of Cu_2O , which is consistent to previous reports (Ref: Malkani, A. *et al.*, *ACS Catal.* **2019**, 9, 474–478). However, the effect of GNSs on the catalytic performance of Cu_2O is similar to that of CDFs. Overall, the SEIRA spectra and reactivity results confirm the successful applications of GNSs as alternative substrates of CDFs for ATR-SEIRAS.

The SEIRA spectra and reactivity results indicate that the reaction mechanism of CORR on the surfaces of Cu-based catalysts could be influenced by the presence of Au. Therefore, developing Cu nanoparticle superlattices via self-assembly process is highly desirable to study the reaction mechanism of CORR on the surfaces of Cu-based catalysts. This is something that we are working on right now. We are very happy to share some preliminary results here. We have fabricated Cu superlattice films through self-assembly of Cu nanocubes. Reactivity results show that the FEs of C_{2+} products on Cu nanocube superlattices are different from those on CDFs supported Cu nanocubes (Fig. R2 in the response letter). SEIRA spectra show that the CO adsorptions on Cu nanocube superlattices are different from those on CDFs supported Cu nanocubes (Fig. R3 in the response letter).

Fig. R1 | Production rates of hydrogen, methane, ethane, formate, ethanol, and ethylene on the surfaces of different catalysts in CO-saturated 0.1 M KOH with CH_3I at -0.7 V vs RHE. Grey: carbon paper supported Cu_2O , blue: CDFs supported Cu_2O , orange: GNSs supported Cu_2O , dark blue: CDFs, and dark red: GNSs. The inset shows the Faradaic efficiencies of production rate at -0.5 V.

Fig. R2 | Faradaic efficiencies of hydrogen, methanol, ethanol, acetate, acetone, methane, and ethylene on the surfaces of different catalysts in CO-saturated 0.5 M KHCO₃ at -0.7 V vs RHE. The inset shows the Faradaic efficiencies of methane and ethanol at -0.5 V.

Fig. R3 | Potential-dependent SEIRA spectra on the surfaces of different substrates in CO-saturated 0.5 M KHCO₃. a Cu superlattice films. b CDFs supported Cu nanocubes.

2) “Homogeneous enhancement would be a strong selling point for the presented superlattices over disordered Au substrates. The authors are encouraged to present homogeneous enhancement through spatial investigations.”

Response: We thank the reviewer for the valuable comments. Since the analytical spot size in FTIR is

around 2.5 mm in diameter, it is difficult to achieve spatial investigation using ATR-SEIRAS. The analytical spot size in Raman spectroscopy is in the order of 0.5–10 μm . We employed surface-enhanced Raman spectroscopy (SERS) to probe the homogeneity in the enhancement on the surfaces of GNSs and CDFs, since the enhancements in both IR and Raman are related to the surface plasmon excitation (Ref: Peter J. Larkin, **2010**, *Infrared and Raman Spectroscopy*; Osawa, M. *et al.*, *Surf. Sci. Lett.* **1992**, 262, L118–L122). The Raman spectra collected in CO-saturated 0.5 M KHCO_3 at 0 V vs RHE show the presence of linearly bonded CO (CO_L) at 2106 cm^{-1} and Au-C vibrations at 392 cm^{-1} on the surfaces of GNSs and CDFs, which are consistent to the results in previous reports (Fig. R4 in the response letter) (Refs: Li, H. *et al.*, *JACS Au* **2021**, 1, 362–368; Berisha, A. *et al.*, *Langmuir* **2018**, 34, 11264–11271). It is noted that the Au-C vibration at 427 cm^{-1} is observed on the surfaces of GNSs but not on the surfaces of CDFs. The SERS results show that the CO adsorption and Au-C vibrations are highly reproducible in terms of both peak position and intensity on three different spots of GNSs, which confirms the excellent homogeneous enhancement on the GNSs (Fig. R4a). On the contrary, the SERS results on CDFs show the poor reproducibility of CO adsorption and Au-C vibrations on three different spots (Fig. R4c). The peak areas of CO_L and Au-C vibrations on GNSs and CDFs are shown in Table R1 in the response letter. The relative standard deviations (RSDs) of the peak areas corresponding to CO_L and Au-C vibrations on the surfaces of GNSs are determined to be 11.6%, 7.3%, and 14.0%, respectively. In contrast, the RSDs of the peak areas corresponding to CO_L and Au-C vibrations on the surfaces of CDFs are determined to be 39.3% and 64.3%, respectively. Therefore, SERS results demonstrate the excellent homogeneous enhancement on the surfaces of GNSs.

Fig. R4 | Surface-enhanced Raman spectra on different substrates in CO-saturated 0.5 M KHCO₃ at 0 V vs RHE. **a** GNSs and **c** CDFs. **b** The picture showing the three spots on GNSs: 1, 2, and 3. **d**, **e**, **f** The pictures showing the three spots on CDFs: 1, 2, and 3. The Raman spectra named as 1, 2, and 3 in **a** are collected at the corresponding three spots in **b**, respectively. The Raman spectra named as 1, 2, and 3 in **c** are collected at the corresponding three spots in **d**, **e**, and **f**, respectively. The scale bar in **b**, **d**, **e**, and **f** is 10 μm . The SERS experiments were performed on a DXR3 Raman microscope (Thermo Scientific) equipped with a 633 nm He-Ne laser.

Table R1. The peak areas of CO_L and Au-C vibrations in the Raman spectra collected on the surfaces of CDFs and GNSs in CO-saturated 0.5 M KHCO₃ at 0 V vs RHE.

Substrate		Peak area of CO _L	Peak area of Au-C vibrations	
			427 cm ⁻¹	~392 cm ⁻¹
GNSs	1	9293.431	4163.031	598.584
	2	10575.983	3714.656	480.986
	3	11748.021	4271.197	467.185
Relative standard deviation (RSD)		11.6%	7.3%	14.0%
CDFs	1	2515.503		1573.518
	2	1147.478		979.285
	3	2630.692		3443.522
Relative standard deviation (RSD)		39.3%		64.3%

3) “The definition of REF (equation 1) is not really valid in my opinion, because the intensity of the studied features largely depends on enhancement which is a spatially varying parameter. Without spatial investigations, the REF equation has no real physical meaning in my opinion. It would be better to relate the vibrational strength to some internal standard, like water vibrations or electrolyte species, and then compare the relative enhancement factor.”

Response: We thank the reviewer for the valuable comments. In this work, we introduce REF to show the significantly improved enhancement effect on the GNSs as compared to the CDFs. Indeed, the intensity of the studied features largely depends on enhancement which is a spatially varying parameter. Considering that the analytical spot size in FTIR is around 2.5 mm in diameter and the diameter of Si crystal is 2 cm, we believe that it is reasonable to investigate the average enhancement effect of different substrates using REF because of the multiple reflections within the Si crystal. We agree with the reviewer that it would be better to relate the vibrational strength to some internal standard and then compare the relative enhancement factor. This is something that we are working on right now. We are very happy to share some preliminary results here. We are trying to use ferrocene as the internal standard to investigate the SEIRA effect on different substrates (Fig. R5 in the response letter). There are two bands located at 1475 cm⁻¹ and 1421 cm⁻¹ corresponding to the bending mode of methylene group or the Fe-C stretching mode and the C-C stretching mode of the ferrocene ring, respectively, which are consistent to the results in previous reports (Ref: Rudnev, A. V. *et al.*, *Electrochim. Acta*, **2013**, *107*, 33–34). The coverage of ferrocene on the surfaces

of substrates could be determined by measuring the effective charges of ferrocene via the cyclic voltammograms (Fig. R5b in the response letter).

Fig. R5 | Potential-dependent SEIRA spectra (a) and cyclic voltammograms of ferrocene (b) on CDFs in Ar-saturated 0.1 M HClO₄.

4) “In Figure 2c, the comparison of the area is speculative if the integrated peaks are not shown. What integration area was used? If a feature is weak but very broad on a large background, the area calculation can still give a large value. Were the spectra background corrected? And for which concentration CPC was this data collected (I am assuming 5 mM, but this should be indicated).”

Response: We thank the reviewer for the valuable comments. The peak areas in Fig. 2c are corresponding to the peaks located at around 2100 cm⁻¹ as shown in Supplementary Fig. 10a and b. The feature of the CO adsorptions is neither too weak nor very broad on a large background. The full widths at half maximum (FWHMs) of the CO adsorptions on the surfaces of CDFs and GNSs are similar to each other. Also, the background of the CO adsorption in the SEIRA spectra are corrected when we were calculating the peak areas using the OMNIC software (Fig. R6 in the response letter). The concentration of CPC for the fabrication of GNSs is 5 mM, which has been added in the caption of Fig. 2c on Page 11 in the revised manuscript. Furthermore, the peak areas of CO adsorptions on CDFs and GNSs in the potential range from 0.9 to -0.3 V are summarized in Supplementary Table 1 in the revised supplementary information.

Action: We have added the concentration of CPC (5 mM) in the caption of Fig. 2c on Page 11 in the revised manuscript.

“c Potential-dependent peak area of adsorbed CO bands on CDFs and GNSs from 0.9 to -0.3 V in 0.1 M HClO₄ (the concentration of CPC for the fabrication of GNSs is 5 mM).”

Also, we have included the peak areas of CO adsorptions on CDFs and GNSs in the potential range from 0.9 to -0.3 V in 0.1 M HClO₄ in Supplementary Table 1 on Page 14 in the revised supplementary information.

Fig. R6 | The background of the CO adsorption in the SEIRA spectra are corrected using the OMNIC software.

Supplementary Table 1. The peak areas of CO adsorptions on CDFs and GNSs in the potential range from 0.9 to -0.3 V in 0.1 M HClO_4 .

Substrate E / V vs RHE	CDFs	GNSs
0.9	0.00145	0.00280
0.8	0.00264	0.00540
0.7	0.00489	0.01170
0.6	0.01140	0.01454
0.5	0.01409	0.02570
0.4	0.02474	0.03350
0.3	0.02552	0.05440
0.2	0.03196	0.07790
0.1	0.02449	0.09990
0	0.01418	0.12470
-0.1	0.00877	0.09570
-0.2	0.00498	0.09080
-0.3	0.00496	0.07270

5) “The changes in the slope of the Stark shift in Figure 2b should be discussed. Why does it deviate from linearity? Why did the authors fit it to a linear fit, while clearly it is not linear? Is the difference of 61 and $63 \text{ cm}^{-1} \text{ V}^{-1}$ then significant?”

Response: We thank the reviewer for the valuable comments. The vibrational energy levels of a molecule are shifted under the influence of an electric field since the molecular dipole moment is different for the molecule in the ground and excited vibrational states. As a result, the molecule exhibits vibrational frequency shifts (vibrational Stark shifts) that increase with the strength of the external electric field. According to previous reports, the Stark shift follows a linear dependence for the sufficiently weak fields (Refs: Bhattacharyya, D. *et al.*, *Chem. Sci.* **2021**, *12*, 10131–10149; Chattopadhyay, A. *et al.*, *J. Am. Chem. Soc.* **1995**, *117*, 1449–1450):

$$\Delta\nu = \nu - \nu_0 = -\Delta\mu F$$

where ν (ν_0) is the frequency of a molecular vibrational mode in the presence (absence) of the field F , and $\Delta\mu$ is the difference in dipole moments (also known as Stark tuning rate) for the molecule in the ground and excited vibrational states. Therefore, the Stark shift was fit to a linear fit in the manuscript. It is worth pointing out that besides external fields such as interfacial electric fields produced at the electrochemical surfaces, other local fields such as the solvation field that originates from solute/solvent interactions also contribute to the vibrational shift (Refs: Bhattacharyya, D. *et al.*, *Chem. Sci.* **2021**, *12*, 10131–10149; Zou, S. *et al.*, *J. Phys. Chem.* **1996**, *100*, 4237–4242; Korzeniewski, C. *et al.*, *J. Chem. Phys.* **1986**, *85*, 4153–4160). The Stark shift deviating from linearity in Fig. 2b could be due to the strong interactions between water molecules and surfactants on the surfaces of GNSs including CPC and CTAB. The relative change of the Stark tuning rates on CDFs and GNSs is 3.3%, therefore, we believe that the difference of 61 and 63 $\text{cm}^{-1} \text{V}^{-1}$ is not significant.

Action: We have added the following sentences when discussing the changes in the slope of the Stark shift in Fig. 2b on Page 9 in the revised manuscript.

“According to previous reports, the Stark shift follows a linear dependence for the sufficiently weak fields:

$$\Delta\nu = \nu - \nu_0 = -\Delta\mu F$$

where ν (ν_0) is the frequency of a molecular vibrational mode in the presence (absence) of the field F , and $\Delta\mu$ is the difference in dipole moments (also known as Stark tuning rate) for the molecule in the ground and excited vibrational states.^{29,30} It is worth pointing out that besides external fields such as interfacial electric fields produced at the electrochemical surfaces, other local fields such as the solvation field that originates from solute/solvent interactions also contribute to the vibrational shift.^{29,31,32} The Stark shift deviating from linearity (Fig. 2b) could be due to the strong interactions between water molecules and surfactants on the surfaces of GNSs including CPC and CTAB.”.

6) “The potentials in Figure 2a are not indicated in the figure, this should be corrected.”

Response: We thank the reviewer for the valuable comments and have added the potentials in Fig. 2a on Page 11 in the revised manuscript.

Action: We have added the potentials in Fig. 2a on Page 11 in the revised manuscript.

Fig. 2 | SEIRA spectra on different films for CO adsorption. **a** Potential-dependent SEIRA spectra on GNSs in CO-saturated 0.1 M HClO₄ solution. **b** Apparent Stark tuning rates of adsorbed CO bands on CDFs and GNSs. **c** Potential-dependent peak area of adsorbed CO bands on CDFs and GNSs from 0.9 to -0.3 V in 0.1 M HClO₄ (the concentration of CPC for the fabrication of GNSs is 5 mM). **d** The SEIRA spectra showing adsorbed CO bands on CDFs and SAFs fabricated in the presence of different concentrations of CPC. The color scheme in **b** applies to **c** and **d**.

7) “Why is no peak observed in Figure 2d for the disordered films with 1 mM CPC? This would almost resemble a CDF right? Why is the peak on CDF observed and not on disordered superlattices?”

Response: We thank the reviewer for the constructive comments. We agree to the reviewer’s opinion that the morphology of disordered films with 1 mM CPC almost resembles that of CDFs. However, the thickness may vary on the disordered films with 1 mM CPC and CDFs. SEM images show that the thickness of disordered gold films with 1 mM CPC and CDFs are around 220 nm and 52 nm, respectively (Fig. R7, a and b in the response letter). According to previous reports, the thickness of metal films on Si crystals for ATR-SEIRAS is around 10–100 nm, because the penetration depth of infrared waves through metal is small (Refs: Li, H. *et al.*, *Chinese J. Catal.* **2022**, *43*, 2772–2791; Ye, J.-Y. *et al.*, *Nano Energy* **2016**, *29*, 414–427).

Therefore, the SEIRA effect on disordered gold films with 1 mM CPC might be much weaker than that on CDFs due to the significantly increased thickness of metal films. FDTD simulations also confirm that the local electromagnetic field is particularly weak on the disordered gold films with 1 mM CPC (Fig. 3). FDTD simulations are further conducted to study the roles of CPC on the SEIRA effect of gold films (Supplementary Fig. 24). It is found that the electromagnetic field strength of GNSs with 5 mM CPC changes little after the modification of CPC, indicating that the contribution of CPC on the SEIRA effect of GNSs is negligible.

Furthermore, the CPC molecules on the surfaces of disordered films could prevent the adsorption of CO onto the surfaces of gold films. Control experiments are conducted to verify the roles of CPC on the SEIRAS measurements. 1 mM CPC solution was utilized to modify the surfaces of chemically deposited gold films (CDFs). Potential-dependent SEIRA spectra on CPC-modified CDFs show that there is no peak in the range from 2000 to 2200 cm^{-1} in Ar-saturated 0.1 M HClO_4 solution and 0.5 M KHCO_3 (Fig. R7, c and e in the response letter), indicating that CPC molecule is stable in the potential range from +1.0 to -0.8 V. Afterwards, CO gas was purged into the electrolytes for 30 min. SEIRA spectra show that there is no peak in the range from 2000 to 2200 cm^{-1} in CO-saturated 0.1 M HClO_4 solution and 0.5 M KHCO_3 , suggesting that CPC prevents the adsorption of CO onto the surfaces of gold films (Fig. R7, d and f in the response letter). Therefore, there is CO adsorption peaks on the surfaces of CDFs but not on the surfaces of disordered self-assembly gold films with 1 mM CPC.

Action: We have conducted FDTD simulations to reveal the roles of CPC on the SEIRA effect of gold films, which is attached as Supplementary Fig. 24 on Page 26 in the revised supplementary information.

Fig. R7 | Typical SEM images of a SAFs (in the presence of 1 mM CPC) and b CDFs. Potential-dependent SEIRA spectra on the surfaces of CPC-modified CDFs (the concentration of CPC is 1 mM) in different electrolytes. c Ar-saturated 0.1 M HClO₄, d CO-saturated 0.1 M HClO₄, e Ar-saturated 0.5 M KHCO₃, and f CO-saturated 0.5 M KHCO₃.

Supplementary Fig. 24 | FDTD simulations of the local electromagnetic field for GNSs before (a, c, and e) and after (b, d, and f) the modification of CPC. Simulated near-field enhancement $|E|^2$ on XY planes (a and b), XZ planes (c and d), and YZ planes (e and f).

8) “When Cu_2O is reduced to Cu, copper itself will give enhancement as well. Is the analysis in Fig 4 corrected for the enhancement of Cu itself? Did the authors perform similar experiments on pure Cu_2O without Au substrates? Can the authors exclude enhancement from Cu, for example through spatial investigations or blank measurements? Or perhaps Raman measurements to study the Au-C and Cu-C vibrations at low Raman shifts?”

Response: We thank the reviewer for the constructive comments. The enhancement effect of Cu is not considered when we are analyzing the results in Fig. 4. Indeed, Cu_2O catalysts would be reduced to Cu during the CORR. Control experiments are conducted to identify the enhancement effect of Cu. 200 μL of Cu_2O ink solution (1 mg mL^{-1}) was placed on the surfaces of Si crystals to produce Cu films for SEIRAS measurements. The amount of Cu_2O was kept to be the same as that for the SEIRAS measurements in Fig.

4. The Cu_2O catalysts cannot be reduced to form continuous Cu films with good conductivity even at a negative potential of -9.3 V vs RHE in Ar-saturated 0.5 M KHCO_3 for 50 min (Fig. R8a in the response letter), which makes it difficult to measure the enhancement effect on Cu_2O without Au substrates using the ATR-SEIRAS setup. Therefore, we decided to measure the enhancement effect on Cu_2O without Au substrates using Raman spectroscopy by studying the Au-C and Cu-C vibrations. GNSs was fabricated on the surfaces of glass carbon electrode and Cu_2O was placed on the top of GNSs (Fig. R8b in the response letter). Platinum wire and Ag/AgCl electrode (3.0 M KCl, Gauss Union) were used as the counter electrode and reference electrode for the Raman measurements, respectively. Before the Raman measurements, the working electrodes were maintained at -0.5 V vs RHE in CO-saturated 0.5 M KHCO_3 for 30 min. The Raman spectra of Cu_2O , GNSs, and GNSs supported Cu_2O were collected at 0 V vs RHE (Fig. R8c in the response letter). It is noted that there are no peaks corresponding to the CO adsorptions and Cu-C bands on the surfaces of Cu_2O . On the contrary, there are several peaks located at 2065 , 356 , and 282 cm^{-1} on GNSs supported Cu_2O , corresponding to CO adsorptions and Cu-C bands on surfaces of Cu. Therefore, Cu films via the reduction of Cu_2O show negligible enhancement effect. The Raman spectra on the surfaces of GNSs show the CO adsorptions and Au-C bands at 2106 , 426 , and 391 cm^{-1} , which further confirms the significant enhancement effect of Au substrates. Furthermore, FDTD calculations are performed to study enhancement effect of Cu_2O catalysts. The strength and distribution of electromagnetic field near the GNSs, GNSs supported Cu_2O , and GNSs supported Cu are similar to each other, which confirms the negligible enhancement effect of Cu_2O and Cu (Fig. R9 in the response letter).

Fig. R8 | a Chronoamperometry profiles for the reduction of Cu_2O film. b Schematic illustration of GNSs supported Cu_2O on glass carbon electrode for the Raman measurements. c Raman spectra on Cu_2O , GNSs, and GNSs supported Cu_2O in CO-saturated 0.5 M KHCO_3 . d-f Images of Cu_2O , GNSs, and GNSs supported Cu_2O . The Raman spectra are collected at the spots labeled by red squares in (d-f). The scale bar in d-f is 10 μm . The SERS experiments were performed on a DXR3 Raman microscope (Thermo Scientific) equipped with a 633 nm He-Ne laser.

Fig. R9 | FDTD simulations of the local electromagnetic field for GNSs (a, d, and g), GNSs supported Cu₂O (b, e, and h), and GNSs supported Cu (c, f, and i). Simulated near-field enhancement $|E|^2$ on XY (a–c), XZ (d–f), and YZ planes (g–i).

9) “What is the Stark tuning rate on Cu? Also, the features in Figure 4 seem to consist of multiple bands (at least two), which are not discussed in detail in the manuscript or deconvoluted. Why are the Stark rates different in different electrolytes (compare Fig. 4c and 4f).”

Response: We thank the reviewer for the valuable comments. Indeed, the features in Fig. 4 consist of several bands, which are consistent to the results in previous reports (Ref: Scarano, D. *et al.*, *Sur. Sci.* **1998**, *411*, 272–285). There are multiple distinct CO adsorption sites located in the region of 2000–2105 cm^{-1} on the surfaces of polycrystalline Cu surfaces, which are attributed to linearly bonded CO (Ref: Malkani, A. *et al.*, *Sci. Adv.* **2020**, *6*, eabd2569). In particular, the two CO adsorption bands that are 15 cm^{-1} apart on Cu surfaces have been assigned to CO bound to step (2089 cm^{-1}) and terrace (2073 cm^{-1}) sites. The bands at 2058 cm^{-1} is attributed to the CO adsorptions on Cu(100) facets. ATR-SEIRAS results in Fig. 4b indicate that the adsorbed CO bands on the surfaces of CDFs supported Cu₂O catalysts shows up at –0.3 V and locates at ~2105 cm^{-1} , which are consistent to the results in previous reports (Refs: Scarano, D. *et al.*, *Sur.*

Sci. **1998**, *411*, 272–285; Zhang, Z. *et al.*, *J. Phys. Chem. C* **2020**, *124*, 21568–21576). The peak position gradually shifts to 2078 cm⁻¹ when the potential decreases to -1.0 V (Fig. 4b). Meanwhile, the adsorbed CO bands appear at the similar position and potential on the surfaces of GNSs supported Cu₂O catalysts (Fig. 4c), which further confirms the successful applications of GNSs as alternative substrates for ATR-SEIRAS. The Stark tuning rates of CO adsorptions on GNSs supported Cu₂O in 0.1 M HClO₄ and 0.5 M KHCO₃ are 14.5 and 62.3 cm⁻¹ V⁻¹, respectively, which are in good agreement with the results in previous reports (Fig. R10 in the response letter) (Refs: Chang, X. *et al.*, *Catal. Sci. Technol.*, **2021**, *11*, 6825–6831; Ooka, H. *et al.*, *Langmuir* **2017**, *33*, 9307–9313).

Besides external fields such as interfacial electric fields produced at the electrochemical surfaces, other local fields such as the solvation field that originates from solute/solvent interactions also contribute to the vibrational shift (Refs: Bhattacharyya, D. *et al.*, *Chem. Sci.* **2021**, *12*, 10131–10149; Zou, S. *et al.*, *J. Phys. Chem.* **1996**, *100*, 4237–4242; Korzeniewski, C. *et al.*, *J. Chem. Phys.* **1986**, *85*, 4153–4160). It is likely that the strong interactions between electrolytes and surfactants on the surfaces of GNSs lead to the different Stark tuning rates in different electrolytes (Refs: Chang, X. *et al.*, *Catal. Sci. Technol.* **2021**, *11*, 6825–6831; Wright *et al.*, *J. Phys. Chem. Lett.* **2022**, *13*, 4905–4911).

Fig. R10 | The apparent Stark tuning rates of adsorbed CO bands on GNSs supported Cu₂O in different electrolytes.

Action: We have added the following sentences when discussing the CO adsorptions in Fig. 4 on Page 16 in the revised manuscript.

“*In situ* ATR-SEIRAS results indicate that the adsorbed CO bands on the surfaces of CDFs supported Cu₂O catalysts show up at -0.3 V and locates at ~2105 cm⁻¹, which are consistent to the results in previous reports.^{42,43} The peak position gradually shifts to 2078 cm⁻¹ when the potential decreases to -1.0 V (Fig. 4b).^{44,45} It is noted that there are multiple distinct CO adsorption sites located in the region of 2000–2105 cm⁻¹, which are attributed to linearly bonded CO.⁴⁶ In particular, the CO adsorption bands could be

assigned to CO bound to step ($\sim 2089\text{ cm}^{-1}$, high wavenumber), terrace sites ($\sim 2073\text{ cm}^{-1}$, main component), and Cu(100) facets ($\sim 2058\text{ cm}^{-1}$, low wavenumber), respectively.”.

10) “The proposed mechanisms in Figure 5 are a bit speculative in my opinion, because the NMR features ascribed to ethanol are quite weak, as well as the corresponding vibrational features in Fig 5d, e for *OCCOH or *CO (Fig. 5c). Are these features confirmed to be C-containing intermediates through ^{13}CO labeling experiments? Different peak positions are ascribed to the *OCCOH intermediates in Figure 5d and e, but not properly discussed. However, these features deconvoluted from the strong OH bending nearby? What shift is expected based on theory, and does this match with the assigned peak positions? I encourage the authors to include more discussion on this dataset, and potentially conduct more experiments to also study the labeling effect on *CO and $^*CH_2/^*CH_3$.”

Response: We thank the reviewer for the valuable comments. To further improve the signal-to-noise ratio of the NMR features, ^{13}C and 1H NMR spectra of liquid products were collected using a Bruker AVIII 400 MHz NMR spectrometer for 10 h with 11000 coadded scans. The ^{13}C and 1H NMR spectra with significantly improved quality are attached as Fig. 5f on Page 21 in the revised manuscript, which clearly show the distinct peaks corresponding to the $^{12}CH_3$ group and $^{13}CH_2$ group in liquid products. The peak with a chemical shift of around 1.05 ppm in the 1H NMR spectra is attributed to $^{12}CH_3$ from CH_3I (Refs: Li, J. *et al.*, *J. Am. Chem. Soc.* **2022**, *144*, 20495–20506). Meanwhile, the peaks at around 3.53 and 3.74 ppm are attributed to $^{13}CH_2$, which is likely from the hydrogenation of ^{13}CO via the CORR. ^{13}C NMR spectra show that there are two peaks at around 17.28 and 55.47 ppm, corresponding to the $^{12}CH_3$ and $^{13}CH_2$ in ethanol, respectively. Therefore, the NMR analysis of the electrolytes after the CORR in the presence of ^{13}CO with CH_3I confirms that ethanol is produced via the coupling of $^*^{13}CO$ and $^*^{12}CH_3$ on the Cu_2O catalysts.

The band located at around 1650 cm^{-1} changes from symmetric on CDFs to asymmetric on GNSs, which can be deconvoluted into two distinct components via the Gaussian fitting (Fig. 5d). The main component located at 1645 cm^{-1} is associated with O-H bending. The low frequency component shifts from 1550 to 1581 cm^{-1} with the applied potential decreasing from -0.2 to -0.7 V , which is attributed to *OCCOH (Refs: Pérez-Gallent, E. *et al.*, *Angew. Chem. Int. Ed.* **2017**, *56*, 3621–3624; Wang, W. *et al.*, *J. Am. Chem. Soc.* **2021**, *143*, 2984–2993; Chang, X. *et al.*, *J. Am. Chem. Soc.* **2020**, *142*, 2975–2983). The peak position of *OCCOH vibration shifts to higher frequency with the decreasing potential, which is likely due to the coverage effect. The redshift of the *OCCOH band from 1581 cm^{-1} in ^{12}CO to 1537 cm^{-1} in ^{13}CO suggests that CO participates in the formation of *OCCOH (Ref: Kim, Y. *et al.*, *Energy Environ. Sci.* **2020**, *13*,

4301–4311), which is an important intermediate to ethanol via the CORR (Fig. 5e). The hypothesis is further supported by the NMR analysis of the electrolytes after the CORR in the presence of ^{13}CO with CH_3I . *In situ* ATR-SEIRAS results indicate that the adsorbed CO bands on the surfaces of CDFs supported Cu_2O catalysts show up at -0.3 V and locates at ~ 2105 cm^{-1} , which are consistent to the results in previous reports (Refs: Zhang, Z. H. *et al.*, *J. Phys. Chem. C* **2020**, *124*, 21568–21576; Scarano, D. *et al.*, *Surf. Sci.* **1998**, *411*, 272–285; Malkani, A. S. *et al.*, *ACS Catal.* **2019**, *9*, 474–478). The peak position gradually shifts to 2078 cm^{-1} when the potential decreases to -1.0 V (Fig. 4b). It is noted that there are multiple distinct CO adsorption sites located in the region of 2000 – 2105 cm^{-1} , which are attributed to linearly bonded CO (Refs: Malkani, A. S. *et al.*, *Sci. Adv.* **2020**, *6*, eabd2569). In particular, the CO adsorption bands could be assigned to CO bound to step (~ 2089 cm^{-1} , high wavenumber), terrace sites (~ 2073 cm^{-1} , main component), and Cu(100) facets (~ 2058 cm^{-1} , low wavenumber), respectively. Meanwhile, the adsorbed CO bands appear at the identical position and potential on the surfaces of GNSs supported Cu_2O catalysts (Fig. 4c). SEIRA spectra in the presence of ^{13}CO -saturated 0.1 M KOH show that there is a peak located at ~ 2000 cm^{-1} , which further confirms the linearly bonded ^{13}CO on the surfaces of GNSs supported Cu_2O (Supplementary Fig. 27, Refs: Wuttig, A. *et al.*, *ACS Cent. Sci.* **2016**, *2*, 522–528; Zhu, S. *et al.*, *J. Am. Chem. Soc.* **2017**, *139*, 15664–15667). The calculated adsorption energy results indicate that CO prefers to adsorb onto the surfaces of Cu(100) and $\text{Cu}_2\text{O}(111)$ facets (Supplementary Fig. 28 and 29). The simulated IR spectra of CO on the surfaces of Cu and Cu_2O are in good agreement with the experimental SEIRA spectra.

There are two peaks located at 2980 and 2959 cm^{-1} after introducing CH_3I into the electrolyte, corresponding to C-H stretching of adsorbed CH_3 from CH_3I on GNSs supported Cu_2O surfaces (Fig. 5a, Refs: Li, J. *et al.*, *J. Am. Chem. Soc.* **2022**, *144*, 20495–20506). Another two bands located at 1263 and 1241 cm^{-1} on GNSs are associated with symmetric and antisymmetric CH_2 twisting modes, respectively (Refs: Liu, G. K. *et al.*, *J. Phys. Chem. C* **2018**, *122*, 21933–21951). It is noted that the intensity of the vibrational peaks corresponding to C-H stretching and CH_2 twisting modes on GNSs decreases with the decreasing potentials. The SEIRA spectra in the presence of CD_3I show that there are two peaks located at 2164 and 2143 cm^{-1} , which are attributed to C-D vibrations on the surfaces of GNSs supported Cu_2O (Supplementary Fig. 31, Refs: Horness, R. E. *et al.*, *J. Am. Chem. Soc.* **2016**, *138*, 1130–1133; Le Sueur, A. L. *et al.*, *J. Am. Chem. Soc.* **2016**, *138*, 7187–7193). SEIRAS results show that the peak positions of C-H stretching and CH_2 twisting modes in the presence of ^{13}CO -saturated 0.1 M KOH (Supplementary Fig. 34) are almost identical to those in Ar-saturated 0.1 M KOH (Fig. 5a), which further confirms the origination of those bands from CH_3I .

Action: We have updated the ^{13}C and ^1H NMR spectra in Fig. 5 on Page 21 in the revised manuscript. Also, we have attached the SEIRAS showing linearly bonded ^{13}CO on the surfaces of GNSs supported Cu_2O as Supplementary Fig. 27 on Page 31 on the revised supplementary information and added the following sentences when discussing the adsorbed CO band on Page 16 in the revised manuscript.

“*In situ* ATR-SEIRAS results indicate that the adsorbed CO bands on the surfaces of CDFs supported Cu_2O catalysts show up at -0.3 V and locates at ~ 2105 cm^{-1} , which are consistent to the results in previous reports.^{42,43} The peak position gradually shifts to 2078 cm^{-1} when the potential decreases to -1.0 V (Fig. 4b).^{44,45} It is noted that there are multiple distinct CO adsorption sites located in the region of 2000 – 2105 cm^{-1} , which are attributed to linearly bonded CO.⁴⁶ In particular, the CO adsorption bands could be assigned to CO bound to step (~ 2089 cm^{-1} , high wavenumber), terrace sites (~ 2073 cm^{-1} , main component), and Cu(100) facets (~ 2058 cm^{-1} , low wavenumber), respectively. Meanwhile, the adsorbed CO bands appear at the identical position and potential on the surfaces of GNSs supported Cu_2O catalysts (Fig. 4c). SEIRA spectra in the presence of ^{13}CO -saturated 0.1 M KOH show that there is a peak located at ~ 2000 cm^{-1} , which further confirms the linearly bonded ^{13}CO on the surfaces of GNSs supported Cu_2O (Supplementary Fig. 27).^{45,47} The calculated adsorption energy results indicate that CO prefers to adsorb onto the surfaces of Cu(100) and $\text{Cu}_2\text{O}(111)$ facets (Supplementary Fig. 28 and 29). The simulated IR spectra of CO on the surfaces of Cu and Cu_2O are in good agreement with the experimental SEIRA spectra.”. We have attached the SEIRAS showing C-D vibrations on the surfaces of GNSs supported Cu_2O as Supplementary Fig. 31 on Page 35 on the revised supplementary information and added the following sentences when discussing the C-H stretching of adsorbed CH_3 from CH_3I on Page 18 in the revised manuscript.

“The SEIRA spectra in the presence of CD_3I show that there are two peaks located at 2164 and 2143 cm^{-1} , which are attributed to C-D vibrations on the surfaces of GNSs supported Cu_2O (Supplementary Fig. 31).^{53,54}”.

We have added the following sentences when discussing the CO adsorptions in the presence of CH_3I on Page 19 in the revised manuscript.

“The weak CO absorption on Cu_2O in the presence of CH_3I is likely due to partially displacement of adsorbed CO by the adsorbed H.⁷”.

Fig. 5 | Potential-dependent SEIRA spectra on the surfaces of different substrates supported Cu₂O in 0.1 M KOH. The C-H stretching and CH₂ twisting modes from CH₃I adsorbed on Cu₂O surfaces in the presence of Ar (**a**) and ¹²CO (**b**). **c** The adsorbed CO, O-H bending, and *OCCOH vibrations in the presence of ¹²CO. The color schemes in **a** and **b** apply to **c**. SEIRA spectra of O-H bending and *OCCOH vibrations on the surfaces of GNSs supported Cu₂O catalysts in (**d**) ¹²CO- and (**e**) ¹³CO-saturated 0.1 M KOH with CH₃I. **f** ¹H NMR and ¹³C NMR peaks corresponding to ethanol produced from the ¹³CORR.

Supplementary Fig. 27 | Potential-dependent SEIRA spectra on the surfaces of GNSs supported Cu_2O in ^{13}CO -saturated 0.1 M KOH solution.

Supplementary Fig. 31 | Potential-dependent SEIRA spectra on the surfaces of GNSs supported Cu₂O in Ar-saturated 0.1 M KOH with CD₃I.

Supplementary Fig. 34 | Potential-dependent SEIRA spectra on GNSs supported Cu₂O catalysts in ¹³CO-saturated 0.1 M KOH with CH₃I.

Reviewer #3

Recommendation:

“This article by Yang et al. reports highly ordered rhombic gold nanocube superlattices (GNSs) as an alternative substrate for operando SEIRAS. It is featured with its high sensitivity, with the vibrations of CO probe molecule being enhanced by 2.4 and 18 times compared to those on traditional chemically deposited Au films in acidic and neutral electrolytes. It is highly recommended to develop such an interesting and powerful method for addressing important spectroscopic issues in complicated CO₂ electroreduction systems. This manuscript for sure has great merit for this field and should be published in Nature Communications. Before acceptance, I would suggest the authors to consider carefully the comments below.”

Response: We thank the reviewer for the positive assessment and constructive suggestions of our work.

1) “As the authors mentioned, in addition to high sensitivity, good reproducibility is also of great importance for a well-recognized experimental method. The SEIRAS experiments using traditional chemically deposited Au films often suffers from bad reproducibility of Au films due to not well controlled preparation procedures. It would be good to show such reproducibility quantitatively in this work.”

Response: We thank the reviewer for the valuable comments. Control experiments are conducted to investigate the reproducibility of CDFs and GNSs for ATR-SEIRAS quantitatively. Three different GNSs via the self-assembly process are named as GNS-1, GNS-2, and GNS-3. SEIRA spectra show that the behaviors of CO adsorption in terms of peak position and intensity are similar to each other on the surfaces of three GNSs (Supplementary Fig. 25 on Page 27 in the revised supplementary information). The electrochemically active surface areas (ECSAs) on GNS-1, GNS-2, and GNS-3 are determined to be 0.34, 0.45, and 0.36 cm², respectively. The maximum peak areas of CO adsorptions, ECSAs, and REFs of GNS-1, GNS-2, and GNS-3 are summarized in Supplementary Table 2 on Page 28 in the revised supplementary information. The RSDs of the maximum peak areas of CO adsorptions, ECSAs, and REFs of GNS-1, GNS-2, and GNS-3 are determined to be 11.57%, 15.29%, and 4.02%, respectively. We have also fabricated three different CDFs via the wet-chemical deposition approach, which are named as CDF-1, CDF-2, and CDF-3, respectively. SEIRA spectra show that the behaviors of CO adsorption vary significantly in terms of the peak intensity on the surfaces of three CDFs (Supplementary Fig. 26 on Page 29 in the revised supplementary information). The electrochemically active surface areas (ECSAs) on CDF-1, CDF-2, and CDF-3 are determined to be 0.93, 0.86, and 0.82 cm², respectively. The maximum peak areas of CO adsorptions, ECSAs, and REFs of CDF-1, CDF-2, and CDF-3 are summarized in Supplementary Table 3 on Page 30 in the revised supplementary information. The RSDs of the maximum peak areas of CO adsorptions, ECSAs, and REFs of CDF-1, CDF-2, and CDF-3 are determined to be 59.19%, 6.40%, and 67.45%, respectively, which are much higher than those on GNSs. Therefore, GNSs prepared via the self-assembly process show significantly improved reproducibility compared to traditional CDFs.

Action: We have added the following sentences on Page 14 in the revised manuscript when discussing the reproducibility of CDFs and GNSs for ATR-SEIRAS.

“Three different GNSs are prepared via the self-assembly process, which are named as GNS-1, GNS-2, and GNS-3. SEIRA spectra show that the behaviors of CO adsorption in terms of peak position and intensity are similar to each other on the surfaces of three GNSs (Supplementary Fig. 25). The electrochemically active surface areas (ECSAs) on GNS-1, GNS-2, and GNS-3 are determined to be 0.34, 0.45, and 0.36 cm², respectively. The maximum peak areas of CO adsorptions, ECSAs, and REFs of GNS-1, GNS-2, and GNS-3 are summarized in Supplementary Table 2. The RSDs of the maximum peak areas of CO adsorptions, ECSAs, and REFs of GNS-1, GNS-2, and GNS-3 are determined to be 11.57%, 15.29%, and 4.02%, respectively. We have also fabricated three different CDFs via the wet-chemical deposition approach, which are named as CDF-1, CDF-2, and CDF-3, respectively. SEIRA spectra show that the behaviors of CO adsorption vary significantly in terms of the peak intensity on the surfaces of three CDFs (Supplementary Fig. 26). The electrochemically active surface areas (ECSAs) on CDF-1, CDF-2, and CDF-3 are determined to be 0.93, 0.86, and 0.82 cm², respectively. The maximum peak areas of CO adsorptions, ECSAs, and REFs of CDF-1, CDF-2, and CDF-3 are summarized in Supplementary Table 3. The RSDs of the maximum

peak areas of CO adsorptions, ECSAs, and REFs of CDF-1, CDF-2, and CDF-3 are determined to be 59.19%, 6.40%, and 67.45%, respectively, which are much higher than those on GNSs. Therefore, GNSs prepared via the self-assembly process show significantly improved reproducibility compared to traditional CDFs.”

We have also attached the SEIRA spectra and electrochemistry results on GNSs and CDFs as Supplementary Fig. 25 and 26 on Page 27 and 29 in the revised supplementary information, respectively. The maximum peak areas of CO adsorptions, ECSAs, and REFs of GNSs and CDFs are summarized in Supplementary Table 2 and Table 3 on Page 28 and 30 in the revised supplementary information, respectively.

Supplementary Fig. 25 | Potential-dependent SEIRA spectra of adsorbed CO bands on three different GNSs in CO-saturated 0.1 M HClO_4 : **a** GNS-1, **b** GNS-2, and **c** GNS-3. Cyclic voltammograms showing the ECSAs of GNSs: **d** GNS-1, **e** GNS-2, and **f** GNS-3.

Supplementary Table 2. The maximum peak areas of CO adsorptions, ECSAs, and REFs of three different GNSs: GNS-1, GNS-2, and GNS-3.

	Maximum peak areas of CO adsorption	ECSAs / cm ²	REFs
GNS-1	0.1003	0.34	0.296
GNS-2	0.1247	0.45	0.274
GNS-3	0.1060	0.36	0.291
Relative standard deviation (RSD)	11.57%	15.29%	4.02%

Supplementary Fig. 26 | Potential-dependent SEIRA spectra of adsorbed CO bands on three different CDFs in CO-saturated 0.1 M HClO₄: **a** CDF-1, **b** CDF-2, and **c** CDF-3. Cyclic voltammograms showing the ECSAs of CDFs: **d** CDF-1, **e** CDF-2, and **f** CDF-3.

Supplementary Table 3. The maximum peak areas of CO adsorptions, ECSAs, and REFs of three different CDFs: CDF-1, CDF-2, and CDF-3.

	Maximum peak areas of CO adsorption	ECSAs / cm ²	REFs
--	-------------------------------------	-------------------------	------

CDF-1	0.1064	0.93	0.144
CDF-2	0.0500	0.86	0.058
CDF-3	0.0348	0.82	0.042
Relative standard deviation (RSD)	59.19%	6.40%	67.45%

2) “It would be also appreciated if the authors comment its advantages over the shell-isolated nanoparticle-enhanced Raman spectroscopy (SHINERS) technique for detecting species during CO₂ electroreduction.”

Response: We thank the reviewer for the valuable comments. Surface-enhanced vibrational spectroscopy including surface-enhanced infrared absorption spectroscopy (SEIRAS) and surface-enhanced Raman spectroscopy (SERS) can provide rich structural information with ultrahigh surface sensitivity, even down to the single-molecule level, which makes it a promising tool for the *in situ* study of catalysis. Generally, SEIRAS and SHINERS provide complementary information, derived from their sensitivity toward the dipole moment and polarizability, respectively, of the substrate. (Ref: Peter J. Larkin, **2010**, Infrared and Raman Spectroscopy). Therefore, both SEIRAS and SERS have been widely utilized to study the CO adsorptions and surface water species during the CO₂ electroreduction. Compared with SEIRA spectroscopy, SHINERS technique is less sensitive to water species, which makes the Kretschmann configuration necessary for SEIRAS. Typically, SEIRAS possesses better signal-to-noise ratios and higher temporal resolutions within its spectral window than SHINERS technique. In this work, highly ordered rhombic GNSs are fabricated as alternative substrates of traditional CDFs for SEIRAS with significantly improved sensitivity and reproducibility (Fig. 2–5 and Supplementary Fig. 25 and 26). Combined with isotopic labeling experiments, the reaction mechanisms for C-C coupling of CORR on Cu-based catalysts are revealed using the GNSs substrates.

Furthermore, the GNSs show excellent homogeneous enhancement because of the highly ordered superlattice structure. Since the analytical spot size in FTIR is around 2.5 mm in diameter, it is difficult to achieve spatial investigation of the homogeneous enhancement using SEIRAS. The analytical spot size in Raman spectroscopy is in the order of 0.5–10 μm. Therefore, Raman spectroscopy is employed to investigate the homogeneous enhancement on the surfaces of GNSs and CDFs. The Raman spectra collected in CO-saturated 0.5 M KHCO₃ at 0 V vs RHE show the presence of linearly bonded CO (CO_L) at 2106 cm⁻¹ and Au-C vibrations at 392 cm⁻¹ on the surfaces of GNSs and CDFs, which are consistent to the results in previous reports (Fig. R4 in the response letter) (Refs: Li *et al.* *JACS Au* **2021**, *1*, 362–368; Berisha *et al.* *Langmuir* **2018**, *34*, 11264–11271). It is noted that the Au-C vibration at 427 cm⁻¹ is observed on the surfaces of GNSs but not on the surfaces of CDFs. The SERS results show that the CO adsorption and Au-C vibrations are highly reproducible in terms of both peak position and intensity on three different spots of GNSs, which confirms the excellent homogeneous enhancement on the GNSs (Fig. R4 in the response letter). On the contrary, the SERS results on CDFs show the poor reproducibility of CO adsorption and Au-C vibrations on three different spots (Fig. R4c in the response letter). The peak areas of CO_L and Au-C vibrations on GNSs and CDFs are shown in Table R1 in the response letter. The relative standard deviations (RSDs) of the peak areas corresponding to CO_L and Au-C vibrations on the surfaces of GNSs are determined to be 11.6%, 7.3%, and 14.0%, respectively. In contrast, the RSDs of the peak areas corresponding to CO_L

and Au-C vibrations on the surfaces of CDFs are determined to be 39.3% and 64.3%, respectively. Therefore, such GNSs can be further applied in SHINERS after coated with ultrathin silica shells with improved homogeneous enhancement effect.

Fig. R4 | Surface-enhanced Raman spectra on different substrates in CO-saturated 0.5 M KHCO₃ at 0 V vs RHE. a GNSs and **c** CDFs. **b** The picture showing the three spots on GNSs: 1, 2, and 3. **d, e, f** The pictures showing the three spots on CDFs: 1, 2, and 3. The Raman spectra named as 1, 2, and 3 in **a** are collected at the corresponding three spots in **b**, respectively. The Raman spectra named as 1, 2, and 3 in **c** are collected at the corresponding three spots in **d, e, and f**, respectively. The scale bar in **b, d, e, and f** is 10 μm. The SERS experiments were performed on a DXR3 Raman microscope (Thermo Scientific) equipped

with a 633 nm He-Ne laser.

Table R1. The peak areas of CO_L and Au-C vibrations in the Raman spectra collected on the surfaces of CDFs and GNSs in CO-saturated 0.5 M KHCO₃ at 0 V vs RHE.

Substrate		Peak area of CO _L	Peak area of Au-C vibrations	
			427 cm ⁻¹	~392 cm ⁻¹
GNSs	1	9293.431	4163.031	598.584
	2	10575.983	3714.656	480.986
	3	11748.021	4271.197	467.185
Relative standard deviation (RSD)		11.6%	7.3%	14.0%
CDFs	1	2515.503		1573.518
	2	1147.478		979.285
	3	2630.692		3443.522
Relative standard deviation (RSD)		39.3%		64.3%

3) “Fig. 1c, it is not clear that why the EIS indicates an excellent conductivity. It is good to also consider the electrode area (with a unit of $\Omega \cdot \text{cm}^2$) and compare to a reference/literature value.”

Response: We thank the reviewer for the valuable comments. The electrochemical impedance spectroscopy (EIS) of GNSs in Fig. 1c has been updated after considering the electrode area on Page 7 in the revised manuscript. Furthermore, the EIS of CDFs has been added as Supplementary Fig. 4 on Page 5 in the revised supplementary information. The Nyquist plots of GNSs and CDFs are fitted using an electrochemical equivalent circuit consisting of three circuit elements, including electrolyte resistance (R_s), and polarization resistance (R_p) in parallel with double layer capacitance (C_{DL}) (Ref: Liu, T. *et al.*, *Chem. Sci.* **2018**, *9*, 4424–4429). The R_s of GNSs is determined to be 78 $\Omega \cdot \text{cm}^2$, which is comparable to that of CDFs and further confirms the excellent conductivity of GNSs (Fig. 1c and Supplementary Fig. 4) (Refs: Kawata, S. *Near-Field Optics and Surface Plasmon Polaritons* Berlin, Heidelberg, 2001; Yang, X. *et al.*, *Angew. Chem. Int. Ed.* **2019**, *58*, 17718–17723).

Action: We have added the following sentences when discussing the conductivity of GNSs on Page 6 and also updated Fig. 1c after considering the electrode area on Page 7 in the revised manuscript. Furthermore, we have added the EIS of CDFs as Supplementary Fig. 4 on Page 5 in the revised supplementary information.

“Moreover, the electrochemical impedance spectroscopy shows the excellent conductivity of GNSs, which makes them suitable for studying electrochemical reactions (Fig. 1c and Supplementary Fig. 4).”

Fig. 1 | The preparation of rhombic gold nanocube superlattices. a Schematic illustration depicting the major steps involved in the formation of SAFs, sedimentation and self-assembly. Highly ordered superlattice films are fabricated by manipulating the Van der Waals force and steric repulsion between Au nanocubes. **b** A typical scanning electron microscopic image of GNSs. The inset present a photo of GNSs with a diameter of 2 cm. **c** Electrochemical impedance spectroscopy of GNSs. **d** Schematic illustration showing near-field enhancement on GNSs.

Supplementary Fig. 4 | Electrochemical impedance spectroscopy of CDFs.

4) “Supplementary Fig. 9, the CO adsorption band appears at up to 2140 cm^{-1} which is almost that of free CO in gas. Is that convincing? CO adsorption on Au also occurs at up to $+0.8 \text{ V}$ vs RHE. However, CO adsorption on Au is usually weak.”

Response: We thank the reviewer for the valuable comments. Indeed, the CO adsorption band in Supplementary Fig. 10 appears at up to 2140 cm^{-1} , which is close to that of free CO in gas. However, the peak position of the bands located at $\sim 2080 \text{ cm}^{-1}$ on the surfaces of CDFs and GNSs shifts to low wavenumber region with the decreasing potential in different electrolytes (Fig. 2b and Supplementary Fig. 10), suggesting that the bands are corresponding to the species specifically adsorbed at the electrochemical interfaces. Considering that the behaviors of the band is similar to those of CO adsorptions on the surfaces of Au (Refs: Blizanac B. B. *et al.*, *J. Am. Chem. Soc.* **2004**, *126*, 10130–10141; Sun S. G. *et al.*, *J. Phys. Chem. B* **1999**, *103*, 2460–2466), we have attributed it to adsorbed CO band on the surfaces of CDFs and GNSs. Control experiments are further conducted to verify whether the band at $\sim 2080 \text{ cm}^{-1}$ is corresponding to free CO in gas. SEIRA spectra show that there are two bands located at 2172 and 2118 cm^{-1} after purging CO for 30 min in the absence of electrolyte, which are attributed to free CO in gas (Refs: Navarro N. M. *et al.*, *J. Phys. Chem. B* **2011**, *115*, 2024–2029; Anibal, D. *et al.*, *Fuller. Nanotub. Car. N.* **2016**, *24*, 225–233; Jia, J. *et al.*, *J. Phys. Chem. B* **2001**, *105*, 3017–3022). It is noted that the bands corresponding to free CO in gas disappear after introducing electrolytes (0.1 M HClO_4 solution) into the spectroelectrochemical cell (Fig. R11a in the response letter). Potential-dependent SEIRA spectra are then collected on the surface of CDFs and there are only specifically adsorbed CO bands on the surfaces of CDFs (Supplementary Fig. 10). Afterwards, we introduce CO gas in path of IR light and the free CO bands in gas appear again. The peak position of the free CO bands in gas stays unchanged at different potentials (Fig. R11b in the response letter), confirming that it is not corresponding to specifically adsorbed CO bands. The totally different behaviors of the bands at $\sim 2080 \text{ cm}^{-1}$ in Supplementary Fig. 10 from those of free CO bands in gas further confirm that they are corresponding to specifically adsorbed CO bands on the surfaces of CDFs and GNSs.

Typically, the CO adsorptions on Au is weak and disappear at +0.6 V vs RHE in 0.5 M NaHCO₃ according to previous reports (Refs: Dunwell, M. *et al.*, *J. Am. Chem. Soc.* **2017**, *139*, 3774–3783). The literature survey reveals that CO adsorptions on Au could appear at up to +0.8 V vs RHE in 0.1 M HClO₄, which is consistent to the SEIRAS results in this work (Refs: Chen, D.-J. *et al.*, *J. Phys. Chem. C* **2016**, *120*, 16132–16139).

Fig. R11 | Time-dependent (a) and potential-dependent (b) SEIRA spectra of the bands corresponding to free CO in gas.

5) “Why is there an obvious shift of CO adsorption bands on CDFs and GNSs at 5 mM?”

Response: We thank the reviewer for the valuable comments. The vibrational energy levels of a molecule are shifted under the influence of an electric field since the molecular dipole moment is different for the molecule in the ground and excited vibrational states. As a result, the molecule exhibits vibrational frequency shifts (vibrational Stark shifts) that increase with the strength of the external electric field (Refs: Bhattacharyya, D. *et al.*, *Chem. Sci.* **2021**, *12*, 10131–10149; Chattopadhyay, A. *et al.*, *J. Am. Chem. Soc.* **1995**, *117*, 1449–1450). Therefore, there is an obvious shift of CO adsorption bands with the change of applied potentials on CDFs and GNSs at 5 mM.

6) “Is there any Cu₂O solution in acid (0.1 M HClO₄ solution)?”

Response: We thank the reviewer for the valuable comments. According to previous reports, metallic Cu catalysts stay stable in acidic electrolytes under the CO₂/CORR conditions (Refs: Huang, J. E. *et al.*, *Science* **2021**, *372*, 1074–1078; Nitopi, S. *et al.*, *Chem. Rev.* **2019**, *119*, 7610–7672; Gu, J. *et al.*, *Nat. Catal.* **2022**,

5, 268–276). To investigate the stability of Cu₂O catalysts in acid (0.1 M HClO₄ solution), the concentration of Cu²⁺ in 0.1 M HClO₄ solution before and after reduction is determined by inductive coupled plasma atomic emission spectroscopy (ICP-AES). The loading of Cu₂O catalysts on the surfaces of GNSs is fixed to be 2.0 mg. ICP-AES results show that 5 ppm Cu²⁺ is dissolved into 20 mL HClO₄ solution within 10 min before reduction (Fig. R12 in the response letter). The fraction of dissolved Cu₂O catalysts is:

$$\frac{\frac{5 \times 20}{1000 \times 63.55} \times (63.55 \times 2 + 16)}{2.0} \times 100\% = 11.26 \text{ wt\%}$$

Meanwhile, ICP-AES results show that 2.1 ppm Cu²⁺ is dissolved into 20 mL HClO₄ solution within 30 min at -0.5 V vs RHE (Fig. R12 in the response letter) and the fraction of dissolved Cu₂O catalysts is:

$$\frac{\frac{2.1 \times 20}{1000 \times 63.55} \times (63.55 \times 2 + 16)}{2.0} \times 100\% = 4.73 \text{ wt\%}$$

Therefore, Cu₂O catalysts are not stable in acidic electrolytes at the open circuit potential (OCP), but stay stable in acidic electrolytes under the CO₂/CORR conditions.

Fig. R12 ICP-AES results of the dissolved Cu₂O in 0.1 M HClO₄ solution before and after reduction. The red curve shows the standard curve for the quantification of Cu²⁺.

7) “Why was C-C coupling of the CORR on Cu₂O catalysts studied in the presence of CH₃I? As the authors mentioned (Fig.6, pathway 4), -CH₃ may be not directly involved in the formation of some products such as ethanol. And, it is not convincing that no signals assigned to -CH₃ are observed using traditional chemically deposited Au film substrate, in a solution containing 0.5 M CH₃I. The concentration of CH₃I should be also high enough on and near the electrode surface.”

Response: We thank the reviewer for the valuable comments. To date, Cu is the only heterogeneous catalyst that has shown a propensity to produce valuable hydrocarbons and alcohols, such as ethylene and ethanol, from the CO₂/CORR. A large number of grain boundaries were found in the Cu catalysts derived from Cu₂O, which contributes to the high selectivity of C₂₊ products due to the undercoordinated sites stabilized by grain boundary surface terminations (Refs: Li, C. W. *et al.*, *Nature* **2014**, *508*, 7497; Ren, D. *et al.*, *ACS Catal.* **2015**, *5*, 2814–2821). Therefore, the C-C coupling of the CORR is studied on Cu₂O catalysts. According to previous reports, the hydrogenation of CO with adsorbed water for the formation of -CH₃ rather than the C-C coupling is the rate-determining step in the formation of C₂₊ products in the CORR (Refs: Chang, X. *et al.*, *Angew. Chem. Int. Ed.* **2022**, *134*, e202111167; Li, J. *et al.*, *J. Am. Chem. Soc.* **2022**, *144*, 20495–20506). Therefore, the mechanistic insights into C-C coupling can be obtained by introducing CH₃I during the CORR, even though -CH₃ is not directly involved in the formation of ethanol through pathway 4 (Fig. 6, Ref: Jiang, K. *et al.*, *Nat. Catal.* **2018**, *1*, 111–119).

According to previous reports (Ref: Dunwell, M. *et al.*, *ACS Catal.* **2018**, *8*, 3399–4008), the concentration of adsorbed species near the electrochemical interfaces might be several magnitudes lower than that in the bulk solution. Therefore, the concentration of -CH₃ near the surfaces of CDFs might be still low even though the bulk concentration of CH₃I is 0.5 M. However, it is difficult to determine the concentration of CH₃I near the electrode surface and further efforts are needed. To investigate whether iodine would affect the C-H stretching modes, SEIRA spectra are collected on the surfaces of CDFs and GNSs supported Cu₂O in 0.1 M KOH with 0.5 M CH₃OH (Fig. R13 in the response letter). It is noted that there are still no peaks in the region of 2800–3000 cm⁻¹ on the surfaces of CDFs supported Cu₂O (Fig. R13a in the response letter), suggesting that the absence of C-H stretching modes in SEIRA spectra is likely due to the relatively low concentration of -CH₃ near the electrochemical interfaces and also the poor SEIRA effect of CDFs. On the contrary, the bands corresponding to C-H stretching modes appear on GNSs supported Cu₂O both in 0.1 M KOH with 0.5 M CH₃OH (Fig. R13b in the response letter), which further confirms the significantly enhanced SEIRA effect on the surfaces of GNSs.

Fig. R13 Potential-dependent SEIRA spectra on the surfaces of CDFs (a) and GNSs (b) supported Cu₂O in 0.1 M KOH with 0.5 M CH₃OH.

REVIEWERS' COMMENTS

Reviewer #2 (Remarks to the Author):

The authors have adequately addressed the questions from me and the other reviewers in my opinion, so I can recommend this work for publication

Reviewer #3 (Remarks to the Author):

The authors have addressed my comments and made great efforts to improve the quality of this manuscript. The reported method is likely to address the sensitivity issues of vibrational spectroscopy under complex electrochemical conditions. Therefore, I would recommend its acceptance in Nature Communications as it is.